# FLASHWORLD: HIGH-QUALITY 3D SCENE GENERATION WITHIN SECONDS

**Xinyang Li**[1*], **Tengfei Wang**[2], **Zixiao Gu**[3], **Shengchuan Zhang**[1], **Chunchao Guo**[2], **Liujuan Cao**[1†]

[1] Key Laboratory of Multimedia Trusted Perception and Efficient Computing, Ministry of Education of China, Xiamen University, [2] Tencent, [3] Yes Lab, Fudan University

## ABSTRACT

We propose `FlashWorld`, a generative model that produces 3D scenes from a single image or text prompt in seconds, $10 \sim 100\times$ faster than previous works while possessing superior rendering quality. Our approach shifts from the conventional multi-view-oriented *(MV-oriented)* paradigm, which generates multi-view images for subsequent 3D reconstruction, to a *3D-oriented* approach where the model directly produces 3D Gaussian representations during multi-view generation. While ensuring 3D consistency, 3D-oriented method typically suffers poor visual quality. `FlashWorld` includes a dual-mode pre-training phase followed by a cross-mode post-training phase, effectively integrating the strengths of both paradigms. Specifically, leveraging the prior from a video diffusion model, we first pre-train a dual-mode multi-view diffusion model, which jointly supports MV-oriented and 3D-oriented generation modes. To bridge the quality gap in 3D-oriented generation, we further propose a cross-mode post-training distillation by matching distribution from consistent 3D-oriented mode to high-quality MV-oriented mode. This not only enhances visual quality while maintaining 3D consistency, but also reduces the required denoising steps for inference. Also, we propose a strategy to leverage massive single-view images and text prompts during this process to enhance the model's generalization to out-of-distribution inputs. Extensive experiments demonstrate the superiority and efficiency of our method. Our code is released at https://github.com/imlixinyang/FlashWorld.

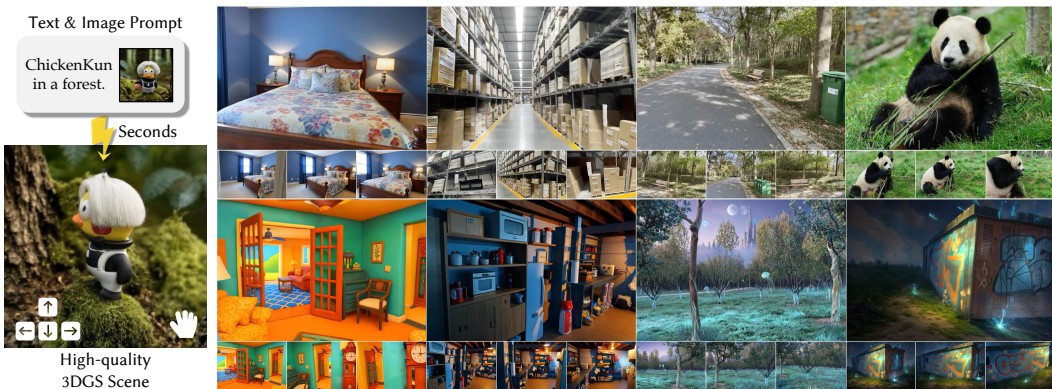

Figure 1: `FlashWorld` enables fast and high-quality 3D scene generation across diverse scenes.

# 1 INTRODUCTION

3D generation shows great promise for applications in gaming, robotics, and VR/AR. However, generating full 3D scenes remains a significant challenge for both quality and efficiency, compared to generating individual 3D objects. These challenges stem from two core obstacles: the scarcity of high-quality 3D scene data and the exponential complexity of modeling real-world scenes.

---

[*]Work done during an internship at Tencent.
[†]Corresponding Authors.

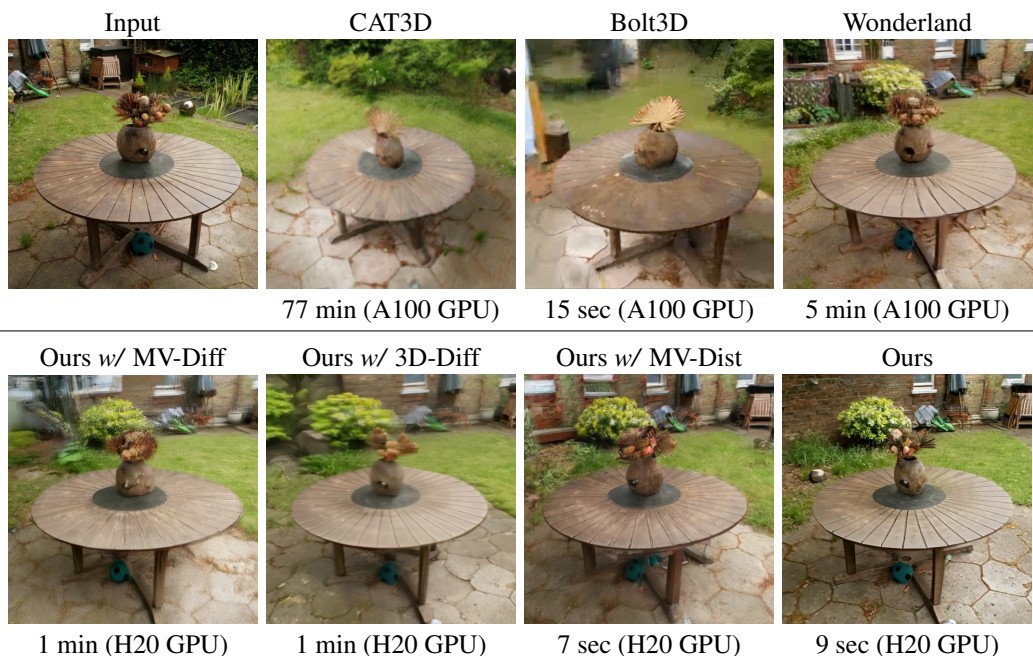

| Input | CAT3D | Bolt3D | Wonderland |
|---|---|---|---|
| | 77 min (A100 GPU) | 15 sec (A100 GPU) | 5 min (A100 GPU) |

| Ours *w/* MV-Diff | Ours *w/* 3D-Diff | Ours *w/* MV-Dist | Ours |
|---|---|---|---|
| 1 min (H20 GPU) | 1 min (H20 GPU) | 7 sec (H20 GPU) | 9 sec (H20 GPU) |

Figure 2: **A brief comparison of different 3D scene generation methods.** MV-oriented diffusion methods (*i.e.*, CAT3D (Gao et al., 2024), Bolt3D (Szymanowicz et al., 2025), Wonderland (Liang et al., 2025), and Ours *w/* MV-Diff) suffer from noisy textures due to multi-view inconsistency. MV-oriented distillation further exacerbates this flaw (*i.e.*, Ours *w/* MV-Dist). 3D-oriented diffusion methods (*i.e.*, Ours *w/* 3D-Diff) suffer from blurry visual effect. Our cross-mode distillation model (*i.e.*, Ours) simultaneously solves these, making the quality of the novel view close to the input view. The time cost per scene, tested on a single GPU, is presented at the bottom of each method.

Early methods typically relied on assembling pre-existing 3D assets (Xu et al., 2002; Yu et al., 2011; Wu et al., 2018; Feng et al., 2023; Çelen et al., 2024; Yang et al., 2024c; Deng et al., 2025) or iteratively reconstructing scenes from inpainted images and depth maps (Cai et al., 2023; Fridman et al., 2023; Höllein et al., 2023; Lei et al., 2023; Yu et al., 2024a; Zhang et al., 2024a;b; Chung et al., 2023; Yu et al., 2025; Shriram et al., 2025; Ni et al., 2025). Yet, without holistic scene-level understanding or multi-view consistency constraints, these approaches often struggle to produce semantically coherent and visually realistic scenes. To address this, scalable data-driven approaches have emerged. The dominant paradigm is a two-stage, multi-view-oriented (MV-oriented) pipeline (Gao et al., 2024; Sun et al., 2024; Wallingford et al., 2024; Zhao et al., 2025; Szymanowicz et al., 2025; Yang et al., 2025; Go et al., 2025a;b): a diffusion model first generates multiple views from text or reference images, and then a 3D reconstruction is performed. However, the lack of explicit 3D constraints during view synthesis often causes geometric and semantic inconsistencies in generated views, leading to a noticeable visual quality gap between synthesized views and the reconstructed 3D scene. Moreover, the considerable computational overhead of both the diffusion and reconstruction stages leads to generation latencies of several minutes to hours, as shown in Fig. 2. These limitations compromise the effectiveness and efficiency of current 3D scene generation methods, blocking their applications.

One promising but relatively less explored direction is the 3D-oriented scene generation pipeline (Xu et al., 2023; Li et al., 2024a;b; Tang et al., 2025; Cai et al., 2024; Zuo et al., 2024). These methods combine differentiable rendering (Mildenhall et al., 2020; Wang et al., 2021; Kerbl et al., 2023) with diffusion models, allowing for direct 3D scene generation without additional reconstruction. However, these generated 3D scenes often suffer from visual artifacts and blurry content. Consequently, they often require an additional refinement stage, which significantly degrades generation efficiency.

To enhance efficiency of diffusion models, post-training distillation techniques, such as consistency model distillation (Song et al., 2023) and distribution matching distillation (Yin et al., 2024b;a; Xie et al., 2024), are often used. However, directly applying distillation amplifies each framework's inherent limitations: *e.g.*, it exacerbates multi-view inconsistency in the MV-oriented pipeline.

In this work, we introduce a novel framework that combines the strengths of both paradigms through distillation, achieving substantial gains in 3D consistency and visual fidelity while significantly accelerating inference speed. Our contributions are briefly summarized as follows:

• We introduce a dual mode pre-training strategy built on a video diffusion model to train a multi-view diffusion model capable of operating in both MV-oriented and 3D-oriented modes.

• We propose a cross-mode post-training strategy, where the MV-oriented mode serves as the teacher to improve visual quality, while the 3D-oriented mode acts as the student to ensure 3D consistency.

• To improve out-of-distribution generalization ability, we introduce a novel strategy that can leverage massive unlabeled image data and text prompts with randomly simulated camera trajectories during post-training, enhancing the model's adaptability to diverse inputs, as shown in Fig. 1.

## 2 PRELIMINARY

**Diffusion models** (Ho et al., 2020) generate data by progressively transforming samples from a standard Gaussian distribution $p(x_T) \sim \mathcal{N}(\mathbf{0}, \mathbf{I})$ into samples from a target data distribution $p(x)$, which have been widely applied across multiple domains, including image synthesis (Rombach et al., 2022), multi-view generation (Shi et al., 2023; Tang et al., 2023), video generation (Blattmann et al., 2023; Wan et al., 2025), and panoramic 3D scenes (HunyuanWorld, 2025). The core methodology involves training a denoising network with optimizable parameters to reconstruct the original data by removing the injected Gaussian noise $\epsilon$ from $x$ according to a predefined noise schedule. The forward process is formulated as: $x_t = F(x, t) = \alpha_t x + \sigma_t \epsilon$, where $\alpha_t$ and $\sigma_t$ jointly control the signal-to-noise ratio at each timestep $t$. The denoising network can be trained to predict clean data $\hat{x}$ from noisy input $x_t$ by minimizing the following objective:

$$\mathcal{L} = \mathbb{E}_{x,t,\epsilon} \left[ \|x - \hat{x}_\theta(x_t, t)\|^2 \right]. \tag{1}$$

Alternative training objectives include predicting noise $\epsilon$ (Ho et al., 2020) or a linear combination of $x_0$ and $\epsilon$, known as $v$-prediction (Salimans & Ho, 2022). All predictions can be converted to the denoised estimate $\mu(x_t, t)$ and represent the gradient of the log probability of the distribution:

$$s(x_t, t) = \nabla_{x_t} \log p_t(x_t) = -\frac{x_t - \alpha_t \mu(x_t, t)}{\sigma_t^2}. \tag{2}$$

**Distribution matching distillation (DMD)** (Yin et al., 2024b;a) is an advanced technique designed to distill a slow, multi-step teacher diffusion model into a fast, few-step student model with comparable generation capabilities. The key component is to minimize the approximate KL divergence across randomly sampled timesteps $t$ and noise inputs $z$ between the smoothed real data distribution $p_{\text{real}}(x_t)$ and the student generator's output distribution $p_{\text{fake}}(x_t)$ by:

$$\nabla \mathcal{L}_{\text{DMD}} = -\mathbb{E}_t \left( \int \left( s_{\text{real}}(F(G_\theta(z), t), t) - s_{\text{fake}}(F(G_\theta(z), t), t) \right) \frac{dG_\theta(z)}{d\theta} dz \right), \tag{3}$$

where $s_{\text{real}}$ and $s_{\text{fake}}$ are approximated scores using diffusion models $\mu_{\text{real}}$ and $\mu_{\text{fake}}$ trained on their respective distributions (Eq. 1). DMD uses a frozen pre-trained diffusion model $\mu_{\text{real}}$ as the teacher, and dynamically updates $\mu_{\text{fake}}$ while training $G_\theta$, using diffusion loss on samples from the generator.

## 3 METHOD

The core of our framework lies in leveraging DMD to transfer knowledge from a MV-oriented multi-view diffusion model, one well-established for high visual quality, to a 3D-oriented few-step multi-view generator, which is inherently endowed with 3D consistency. However, this paradigm introduces two key challenges: First, for open-world 3D scene generation, the 3D-oriented few-step generator requires a sufficiently robust prior and strong generative capacity from the start. Without this, the training process is prone to collapse. Second, due to the limited quantity and diversity of high-quality multi-view datasets, it becomes critical to develop a strategy that effectively handles scenarios with diverse styles, object categories, and camera trajectories. Specifically, To address these challenges, we first design a dual-mode pre-training strategy as detailed in Sec. 3.1. This strategy

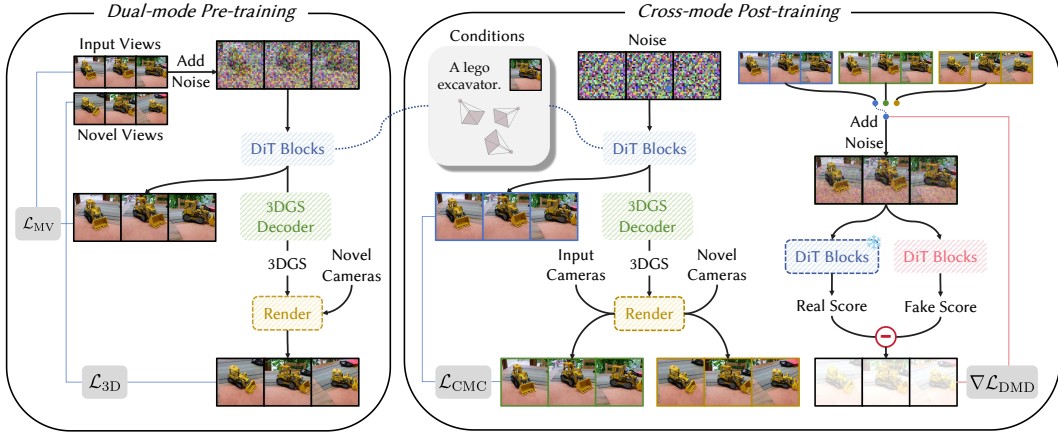

Figure 3: **Method overview.** We first pre-train a dual-mode multi-view latent diffusion model using multi-view datasets, and then employ an cross-mode distillation post-training strategy to accelerate generation while enhancing visual quality and inheriting 3D consistency.

yields a multi-view diffusion model that operates in two distinct modes: a MV-oriented mode for high visual fidelity and a 3D-oriented mode for inherent 3D consistency. Subsequently, in Sec. 3.2, we present a cross-mode post-training framework to bridge these two modes: the MV-oriented mode acts as the *teacher*, supplying score distillation gradients to ensure visual quality; the 3D-oriented mode serves as the *student*, learning to inherit the teacher's distribution while preserving 3D consistency. Furthermore, to explicitly tackle out-of-distribution generalization, in Sec. 3.3, we introduce a strategy that can leverage single-view image data, text prompts, and pre-defined camera trajectories, boosting the model's adaptability to diverse scenarios.

## 3.1 Dual-mode Pre-training

In this stage, we pre-train a dual-mode multi-view latent diffusion model using multi-view datasets, as illustrated in Fig. 3 (left). For each training iteration, we sample a batch containing multi-view images $\mathcal{X}$, their corresponding camera parameters $\mathcal{C}$, and additional conditioning information $y$ (such as a text prompt or a single-view image). The multi-view images are first encoded into the latent space to obtain multi-view latents $\mathcal{Z} = E(\mathcal{X})$. A forward diffusion process is then applied to produce noisy multi-view latents $\mathcal{Z}_t = \alpha_t \mathcal{Z} + \sigma_t \epsilon$ at a randomly sampled timestep $t$.

The noisy latents $\mathcal{Z}_t$, together with the camera parameters $\mathcal{C}$ and conditioning $y$, are input to the denoising network for reverse denoising training. We represent cameras using Reference-Point Plücker Coordinates (Cai et al., 2024) raymaps. The denoising network is a Diffusion Transformer (DiT) (Peebles & Xie, 2023) enhanced with 3D attention blocks, and outputs both a denoised estimate $\hat{\mathcal{Z}}_{\text{MV}}$ and an auxiliary multi-view feature $\mathcal{F}$. For the MV-oriented mode, we optimize:

$$\mathcal{L}_{\text{MV}} = \mathbb{E}_{\mathcal{X}, t, \epsilon, y, \mathcal{C}} \left[ \left\| \mathcal{Z} - \hat{\mathcal{Z}}_{\text{MV}} \right\|^2 \right]. \tag{4}$$

To enable 3D-oriented generation, we decode 3D Gaussian parameters from the multi-view feature $\mathcal{F}$ using a 3DGS decoder: $\{\tau, \boldsymbol{q}, \boldsymbol{s}, \alpha, \boldsymbol{c}\} = D_{\mathcal{G}}(\mathcal{F})$, where $\tau, \boldsymbol{q}, \boldsymbol{s}, \alpha$, and $\boldsymbol{c}$ represent the depth, rotation quaternion, scale, opacity, and spherical harmonics coefficients of the 3D Gaussians, respectively. The 3DGS decoder $D_{\mathcal{G}}$ is initialized from the original latent decoder $D$, with its first and last convolutional layers re-initialized to accommodate the additional features and output channels required for the Gaussian parameters. The predicted depth is then converted to pixel-aligned Gaussian points via $\boldsymbol{\mu} = \boldsymbol{o} + \tau \boldsymbol{d}$, where $\boldsymbol{o}$ and $\boldsymbol{d}$ denote the camera origin and ray direction, respectively. For the 3D-oriented mode, we optimize the following loss:

$$\mathcal{L}_{\text{3D}} = \mathbb{E}_{\mathcal{X}, t, \epsilon, y, \mathcal{C}} \left[ \| \mathcal{X}_{\text{novel}} - R(\mathcal{G}, \mathcal{C}_{\text{novel}}) \|^2 \right], \tag{5}$$

where $R$ denotes the rendering operation, $\mathcal{G} = \{\boldsymbol{\mu}, \boldsymbol{q}, \boldsymbol{s}, \alpha, \boldsymbol{c}\}$ is the set of 3D Gaussians, and $\mathcal{X}_{\text{novel}}$, $\mathcal{C}_{\text{novel}}$ are the ground-truth novel-view images and their associated cameras, respectively. During

inference, both MV-oriented and 3D-oriented modes can be used for denoising (Li et al., 2024a;b). In particular, for the 3D-oriented mode, the model predicts the estimated clean multi-view latents as $\hat{\mathcal{Z}}_{3D} = E(R(\mathcal{G}, \mathcal{C}))$.

In contrast to previous methods (Li et al., 2024a;b) that are initialized from image diffusion models (Rombach et al., 2022), we initialize our framework with a video diffusion model (Wan et al., 2025). We observe that this video model not only converges more rapidly, but also features a powerful VAE with a higher compression rate, enabling support for a larger number of views (*i.e.*, 24) and higher output resolutions (*i.e.*, 480P).

## 3.2 CROSS-MODE POST-TRAINING

After pre-training, we employ an asymmetric distillation strategy to accelerate generation while enhancing visual quality and inheriting 3D consistency, as shown in Fig 3 (right). Specifically, we observe that while the MV-oriented mode exhibits poor consistency, it can generate multi-view images with high visual quality; thus, we leverage the MV-oriented mode of our dual-mode multi-view latent diffusion model as the real teacher $\mu_{\text{real}}$: this teacher model is frozen, tasked with computing the real score gradient. Another copy of the model $\mu_{\text{fake}}$ is dynamically updated to estimate the fake score corresponding to the current distribution of the distilled generator. Meanwhile, our few-step student model is initialized with the 3D-oriented mode of our dual-mode multi-view latent diffusion model.

The 3D-oriented multi-view generation process alternates between denoising and noise injection steps to enhance sample quality of the 3D scenes following LCMs (Luo et al., 2023). Specifically, we first define a schedule of $N$ timesteps, denoted as $\{t_1, t_2, \cdots, t_N\}$, where $N$ is typically small (*e.g.*, 4). Starting from a randomly sampled noise $\mathcal{Z}_{t_1} = z \sim \mathcal{N}(\mathbf{0}, \mathbf{I})$, we alternate between 3D-oriented denoising updates $\hat{\mathcal{Z}}_{t_i} = E(R(G_{\theta,3D}(\mathcal{Z}_{t_i}, t_i, y, \mathcal{C}), \mathcal{C}))$ and forward diffusion steps $\mathcal{Z}_{t_{i+1}} = \alpha_{t_{i+1}}\hat{\mathcal{Z}}_{t_i} + \sigma_{t_{i+1}}\epsilon$ where $G_{\theta,3D}$ is the 3DGS generator and $\epsilon \sim \mathcal{N}(\mathbf{0}, \mathbf{I})$, until obtaining the 3D Gaussians at the final step (*i.e.*, $G_{\theta,3D}(\mathcal{Z}_{t_N}, t_N, y, \mathcal{C})$). At each step, the multi-view denoising update is performed based on rendering, thereby ensuring that 3D consistency is maintained throughout the process.

During distillation training, we adopt the DMD2 algorithm (Yin et al., 2024a), which includes a DMD objective (*i.e.*, Eq. 3) and a standard non-saturating GAN objective (Goodfellow et al., 2020), where the logits value required by the GAN loss is obtained by adding an extra classification branch with several convolutional layers at the end of the fake score network. We adopt the estimated R1 regularization (Lin et al., 2025a) to stabilize the GAN training. The DMD objective and the GAN objective are employed to optimize both the original and novel views.

We also observe that relying solely on the above strategy can lead to the generation of scenes with unstable floating artifacts. We hypothesize that this instability arises from the challenges in optimizing with noisy gradients introduced by Gaussian rendering and latent encoding. To address this, during post-training, we additionally update an MV-oriented student model at a lower frequency. This model shares the same DiT backbone as the 3D-oriented student model. To encourage alignment between the two modes, we introduce a cross-mode consistency loss:

$$\mathcal{L}_{\text{CMC}} = \mathbb{E}_{z,t,\epsilon,y,\mathcal{C},i} \left[ \lambda \left\| E(R(G_{\theta,3D}(\mathcal{Z}_{t_i}, t_i, y, \mathcal{C}), \mathcal{C})) - G_{\theta,\text{MV}}(\mathcal{Z}_{t_i}, t_i, y, \mathcal{C}) \right\|^2 \right], \qquad (6)$$

where $\lambda$ is a small weighting factor (*i.e.*, 0.1). Because the MV-oriented mode prediction are less affected by unstable rendering gradients, this consistency loss regularizes the 3D-oriented mode to produce more stable and reliable generations.

## 3.3 OUT-OF-DISTRIBUTION DATA CO-TRAINING.

During pre-training, it is common to jointly train on image and video generation tasks to enhance the model's generalization ability. While this approach benefits the DiT backbone, it does not optimize the 3DGS decoder, potentially limiting the range of inputs the 3DGS decoder can effectively process. To address this, in the post-training phase, we introduce a strategy to broaden the model's input distribution and improve generalization to diverse scenes, even when multi-view data is limited in quantity and variety. Specifically, we combine image or text conditions sampled from image datasets with random camera trajectories, which can be drawn either from multi-view sequences or from a

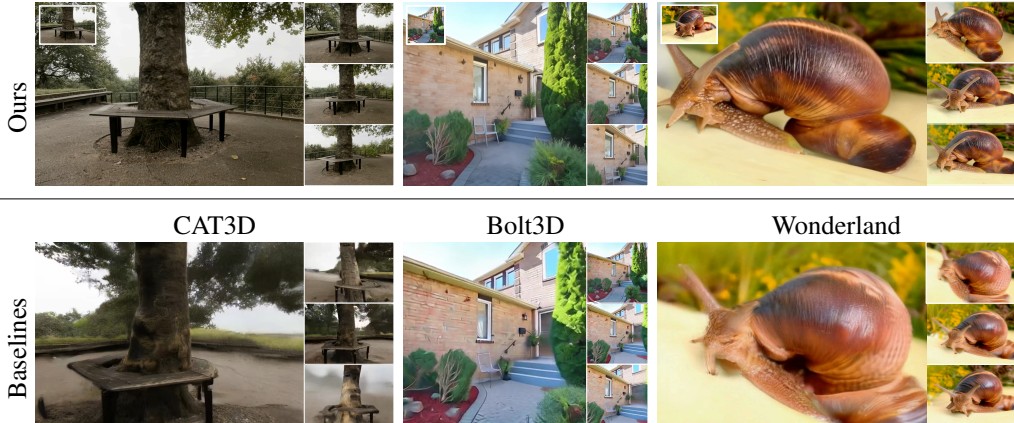

Figure 4: **Image-to-3D scene generation results of different methods**.

set of predefined trajectories. Importantly, we omit the GAN loss during this co-training process to prevent distribution mismatches. This approach not only enhances the model's generalization to a wide range of input images and text prompts, but also increases its robustness when encountering out-of-distribution camera trajectories. The details of this strategy are provided in Appendix A .

## 4 EXPERIMENTS

In this section, we evaluate the performance of our method on various benchmarks, including image-to-3D scene generation, text-to-3D scene generation, and WorldScore benchmark. For implementation details, please refer to Appendix A.

### 4.1 COMPARISON ON IMAGE-TO-3D SCENE GENERATION

We present a qualitative comparison with state-of-the-art image-to-3D scene generation methods in Fig. 4. These baselines are MV-oriented, including: CAT3D (Gao et al., 2024), which generates novel views via multi-view diffusion followed by optimization-based 3D reconstruction; Bolt3D (Szymanowicz et al., 2025), which synthesizes both appearance and geometry for novel views and then applies a feed-forward 3D reconstruction; and Wonderland (Liang et al., 2025), a leading approach that leverages a powerful video diffusion model and latent-based feed-forward 3D reconstruction. As these methods are not open-sourced, we utilize the video results provided in their respective project pages for visualization. We employ ViPE (Huang et al., 2025) to estimate camera poses and intrinsics from the baseline videos. CAT3D struggles to generate complex scenes, resulting in blurry outputs and missing geometric details. Bolt3D also exhibits inaccurate geometric details, such as imprecise tree branches and needle-like leaves. Wonderland suffers from repeated and distorted Gaussian artifacts, especially under large camera pose changes. Overall, these MV-oriented methods fail to generate complex scenes, primarily due to insufficient multi-view consistency. In contrast, our model produces high-fidelity, detailed scenes and successfully recovers intricate structures (*e.g.*, leaves, iron fences, and tentacles), highlighting the advantages of our 3D-oriented pipeline.

### 4.2 COMPARISON ON TEXT-TO-3D SCENE GENERATION

We compare our method against several state-of-the-art text-to-3D scene generation approaches, including Director3D (Li et al., 2024b), Prometheus (Yang et al., 2025), SplatFlow (Go et al., 2025a), and VideoRFSplat (Go et al., 2025a). A qualitative comparison is presented in Fig. 5. Director3D relies on per-scene refinement, which frequently introduces blurry and wave-like artifacts in the generated results. In contrast, our model produces accurate objects with fine-grained details, such as animal fur, while preserving realistic backgrounds. Prometheus does not utilize refinement, and due to the inherent inconsistency of its MV-oriented pipeline, the generated scenes are often blurry and may exhibit incorrect object geometries (*e.g.*, chair legs). Our approach, however, is

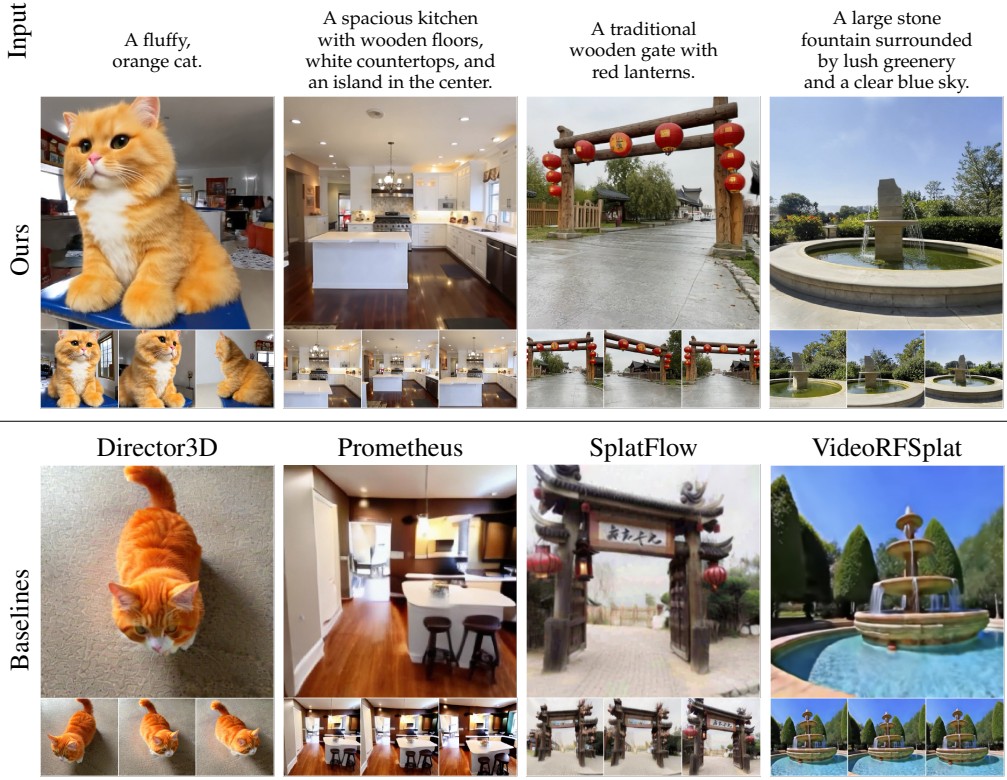

Figure 5: **Text-to-3D scene generation results of different methods**.

| Method | T3Bench-200 | | | | | DL3DV-200 | | | | | WorldScore-200 | | | | | Time Cost |
|--------|-------------|---|---|---|---|-----------|---|---|---|---|----------------|---|---|---|---|------|
| | Q-Align IQA | Q-Align IAA | CLIP IQA+ | CLIP Aesthetic | CLIP Score | Q-Align IQA | Q-Align IAA | CLIP IQA+ | CLIP Aesthetic | CLIP Score | Q-Align IQA | Q-Align IAA | CLIP IQA+ | CLIP Aesthetic | CLIP Score | |
| Director3D | 3.24 | 1.95 | 0.43 | 4.70 | 27.84 | 2.51 | 1.78 | 0.34 | 4.55 | 26.12 | 2.55 | 2.47 | 0.35 | 5.32 | 29.05 | 7 min |
| Promtheus | 2.34 | 1.92 | 0.34 | 4.76 | 24.85 | 2.07 | 1.99 | 0.35 | 4.69 | 23.49 | 2.45 | 2.94 | 0.37 | 5.65 | 28.07 | 15 sec |
| Ours | 4.12 | 2.26 | 0.54 | 4.49 | 27.68 | 3.96 | 2.27 | 0.50 | 4.77 | 27.63 | 3.76 | 2.55 | 0.49 | 5.08 | 29.13 | 9 sec |

Table 1: **Quantitative comparison on text-to-3D scene generation.** Cell background colors indicate the method is the best , second best , or third best on this metric.

capable of generating structurally rich and precise objects in complex scenes, even under large camera movements. SplatFlow and VideoRFSplat also suffer from blurry artifacts and have difficulty reproducing fine details, such as those found in floors and grass. In comparison, our model generates realistic details while maintaining semantic consistency with the input text prompt.

We further perform a comprehensive quantitative evaluation for this task. Specifically, we sample 600 text prompts from T3Bench (He et al., 2023), DL3DV (Ling et al., 2024), and WorldScore (Duan et al., 2025), covering object-centric and general scenes. As all compared methods are based on 3D Gaussian representations, metrics related to camera control and 3D consistency are not applicable in this setting. Accordingly, we concentrate on the quality evaluation metrics utilized, including CLIP IQA+ (Wang et al., 2023), CLIP Aesthetic (Schuhmann, 2022), the text-image alignment score (CLIP Score) (Hessel et al., 2021), as well as the latest LMM-based Q-Align (Wu et al., 2024) image quality metric. The quantitative results are summarized in Tab. 1. It is evident that our model achieves superior performance on the majority of quality evaluation metrics. For CLIP-Aesthetic, we note that this metric sometimes favor smooth outputs, which may not always align with the detailed and realistic results produced by our method. Our method also attains the highest CLIP Score for two subsets, demonstrating the strong text alignment ability of our method. In addition, we report the average time required to generate a single scene for each method on a single H20 GPU. Our method demonstrates a substantial speed advantage over other approaches. Remarkably, this efficiency is

Ours                                    WonderWorld

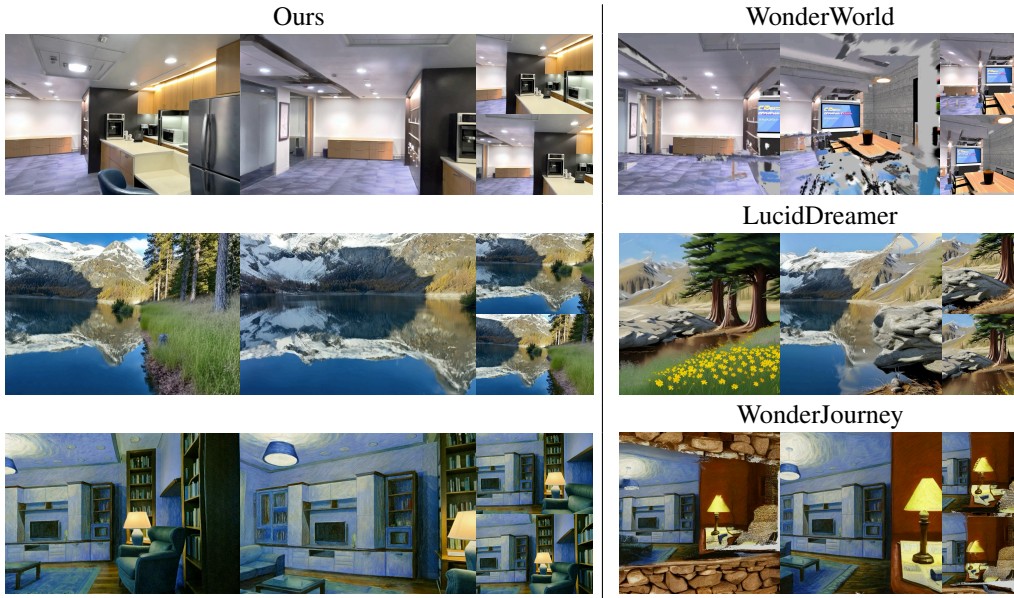

LucidDreamer

WonderJourney

Figure 6: **3D scene generation results of different methods on WorldScore benchmark.**

| Method | 3D Consistency | Photometric Consistency | Object Control | Content Alignment | Style Consistency | Subjective Quality | Average | Time Cost |
|---|---|---|---|---|---|---|---|---|
| WonderJourney | 80.60 | 79.03 | 34.81 | 38.37 | 67.52 | **61.49** | 60.30 | 6 min |
| LucidDreamer | **90.37** | **90.20** | 43.48 | **59.41** | 66.41 | 48.02 | 66.32 | 6 min |
| WonderWorld | 86.91 | 85.56 | **52.09** | 56.82 | 75.92 | 41.28 | 66.43 | 10 sec |
| Ours | 85.87 | 86.72 | 49.61 | 53.96 | **81.52** | 54.63 | **68.72** | **9 sec** |

Table 2: **Quantitative comparison on WorldScore benchmark.** Note that the time cost of the baselines is tested on $1\times$ H100 GPU, while our time cost is tested on $1\times$ H20 GPU.

maintained even when our method produces results with higher resolution and a greater number of frames. In addition, our approach leverages a unified model that seamlessly handles both image-to-3D and text-to-3D tasks without requiring separate training processes. This unified framework not only simplifies the overall workflow but also substantially reduces the training cost.

### 4.3 COMPARISON ON WORLDSCORE BENCHMARK

We further conduct a comprehensive evaluation on the recent WorldScore (Duan et al., 2025) benchmark. The static subset of WorldScore comprises 2,000 test examples, encompassing a diverse array of worlds with varying styles, scenarios, and objects. Each test case provides an input image, a text prompt, and a camera trajectory as conditions for generation. The evaluation protocol is designed to assess two primary aspects of world generation: controllability and quality. For baselines, we select three state-of-the-art 3D generation methods: WonderJourney (Yu et al., 2024a), which iteratively completes novel-view images and depth maps based on point clouds; LucidDreamer (Chung et al., 2023), which also performs iterative novel view completion but utilizes 3DGS for rendering; and WonderWorld (Yu et al., 2025), which improves generation quality through the use of layered Gaussian surfels. Since our comparison focuses exclusively on 3D generation methods, the "Camera Control" metric primarily reflects the robustness of the evaluation protocol for each method, and is thus less informative in this context. Accordingly, we omit this metric from our comparison. Additionally, the original WorldScore benchmark evaluates most metrics only on anchor frames, which is suboptimal for 3D world generation tasks that require novel view synthesis. To ensure a fairer comparison, we re-evaluate these metrics by randomly sampled frames within specific intervals. Qualitative and quantitative comparisons are shown in Fig. 6 and Tab. 2, respectively. Our method achieves the highest average score and the fastest inference speed among all compared approaches. In particular, our model achieves the best "Style Consistency" and secures the second place in "Pho-

| Config. | T3Bench-200 | | | | | DL3DV-200 | | | | | WorldScore-200 | | | | |
| --- | --- | --- | --- | --- | --- | --- | --- | --- | --- | --- | --- | --- | --- | --- | --- |
| | Q-Align IQA | Q-Align IAA | CLIP IQA+ | CLIP Aesthetic | CLIP Score | Q-Align IQA | Q-Align IAA | CLIP IQA+ | CLIP Aesthetic | CLIP Score | Q-Align IQA | Q-Align IAA | CLIP IQA+ | CLIP Aesthetic | CLIP Score |
| A | 3.11 | 2.03 | 0.41 | 4.36 | 25.34 | 2.64 | 2.09 | 0.39 | 4.60 | 24.49 | 2.48 | 2.10 | 0.35 | 4.78 | 27.40 |
| B | 2.61 | 1.68 | 0.37 | 4.11 | 22.92 | 2.71 | 1.96 | 0.40 | 4.54 | 22.71 | 2.74 | 2.16 | 0.33 | 4.83 | 26.11 |
| C | 3.46 | 2.12 | 0.45 | 4.42 | 26.95 | 2.99 | 2.05 | 0.42 | 4.57 | 26.41 | 3.06 | 2.18 | 0.42 | 4.92 | 28.71 |
| D | 4.12 | 2.31 | 0.52 | 4.52 | 27.59 | 4.02 | 2.35 | 0.51 | 4.80 | 27.90 | 3.90 | 2.71 | 0.51 | 5.12 | 29.09 |
| E | 3.98 | 2.50 | 0.53 | 4.58 | 27.04 | 3.89 | 2.35 | 0.50 | 4.82 | 27.45 | 3.66 | 2.56 | 0.47 | 4.95 | 28.76 |
| F | 4.12 | 2.26 | 0.54 | 4.49 | 27.68 | 3.96 | 2.27 | 0.50 | 4.77 | 27.63 | 3.76 | 2.55 | 0.49 | 5.08 | 29.13 |

Table 3: **Quantitative ablation studies.** The letters A–F correspond to different model variants: **(A)** *w/* MV-Diff, **(B)** *w/* 3D-Diff, **(C)** *w/* MV-Dist, **(D)** *w/o* CMC, **(E)** *w/o* OOD, and **(F)** Full model.

| *w/* MV-Diff | *w/* 3D-Diff | *w/* MV-Dist | *w/o* CMC | *w/o* OOD | Full model |
| --- | --- | --- | --- | --- | --- |

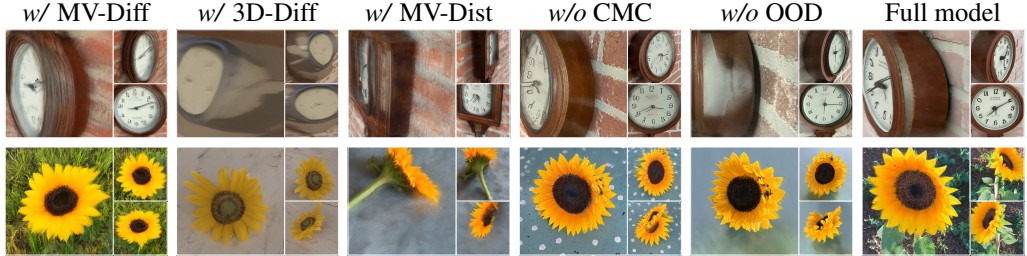

Figure 7: **Qualitative ablation studies.** Prompts: (Top) "A vintage clock hanging on a brick wall"; (Bottom) "A bright sunflower in a field".

tometric Consistency", "Object Control", and "Subjective Quality", reflecting a well-balanced and robust capability across controllability and quality. While our method yields relatively lower scores in "3D Consistency" and "Content Alignment", these results can be attributed to methodological differences: for "3D Consistency", all baselines utilize monocular depth estimation models that are closely aligned with the evaluation protocol, whereas our approach relies solely on RGB supervision without explicit depth guidance; for "Content Alignment", our method does not directly manipulate the anchor frame content, in contrast to the baselines. Qualitative analysis further reveals that baseline methods frequently exhibit unnatural transitions, discontinuous content, and visible holes in the generated scenes, which may not be fully reflected by the current metrics. Overall, our approach demonstrates superior consistency and faithful generation over existing methods.

## 4.4 ABLATION STUDY

In Fig. 2, we show the generation results of various ablation models for image-to-3D scene generation. The outcomes align well with our expectations: both the MV-oriented diffusion model (*w/* MV-Diff) and the MV-oriented distillation model (*w/* MV-Dist) exhibit noisy 3D reconstruction due to multi-view inconsistency, while the 3D-oriented diffusion model (*w/* 3D-Diff) produces blurry visual results. To further validate the effectiveness of each proposed strategy, we conduct more comprehensive ablation studies on text-to-3D scene generation. Quantitative and qualitative results are summarized in Tab. 3 and Fig. 7, respectively. Consistently, the first three ablation models continue to demonstrate worse visual quality and weaker text alignment. The model without cross-mode consistency loss (*w/o* CMC) achieves competitive, and in some cases superior, scores on most quantitative metrics compared to our full model. However, qualitative analysis reveals that this model is susceptible to floating and duplicated artifacts. The model without out-of-distribution data (*w/o* OOD) is more prone to semantic misalignment (*e.g.*, "field") and exhibits a drop in quantitative text alignment metrics. This issue is exacerbated on T3Bench and WorldScore, which differ in distribution from the original multi-view data, highlighting the importance of incorporating OOD data to improve generalization.

## 5 CONCLUSION

We propose an efficient yet powerful model for 3D scene generation, named `FlashWorld`. At the core of our approach is a novel distillation strategy, which transfers high visual fidelity from a multi-view-oriented diffusion model to a 3D-oriented multi-view generative model endowed with perfect 3D consistency. To achieve this, we design a dual-mode pre-training phase and a cross-mode

post-training phase, and introduce an out-of-distribution data co-training strategy to boost the model's generalization. Our method achieves state-of-the-art performance on multiple tasks, while offering significant advantages in inference speed. The efficiency and effectiveness of our approach are well-positioned to advance applications of 3D scene generation. Future work includes incorporating autoregressive generation and extending our framework to dynamic 4D scene generation tasks.

## ACKNOWLEDGMENT

This work was supported by National Science and Technology Major Project (No. 2025YFE0113500), the National Science Fund for Distinguished Young Scholars (No. 62525605), and the National Natural Science Foundation of China (No. U25B2066, No. U22B2051, and No. 62272401).

## ETHICS STATEMENT

`FlashWorld` enable fast, high-quality 3D scene generation, lowering barriers for content creation in fields like gaming, VR/AR, and digital media. This democratization can benefit small studios and independent creators, but also raises concerns about potential misuse (*e.g.*, fake and harmful 3D environments) and dataset bias (*e.g.*, gender and race on human-related subjects). We encourage further research on detection of AI-generated 3D content and careful consideration of ethical implications in real-world applications.

## REPRODUCIBILITY STATEMENT

We are committed to ensuring the reproducibility of all results reported in this paper, in accordance with ICLR standards. To this end, we publicly release our codebase, pre-trained model checkpoints, and all scripts necessary to reproduce our experiments and main results upon publication. Comprehensive details regarding model architecture, training procedures, hyperparameters, and dataset preprocessing are provided in the Appendix A to facilitate independent verification. We also specify all evaluation metrics and protocols used in our experiments. Furthermore, we will provide instructions for environment setup and hardware requirements to enable seamless replication.

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

## A  TRAINING DETAILS

**Architecture configuration.** Our dual-mode multi-view latent diffusion model is initialized with WAN2.2-5B-IT2V (Wan et al., 2025), with the detailed architecture shown in Fig. 8. For both pre-training and post-training, we adopt 24 key frames as input views. The spatial downsampling factor from image space to latent space is set to 16. The auxiliary multi-view feature has a channel dimension of 1024. The discriminator head is a CNN with several residual blocks (He et al., 2016).

**Pre-training configuration.** We set the learning rates for both the transformer and 3DGS decoder to $2 \times 10^{-6}$. We use a weight decay of $1 \times 10^{-6}$ and Adam (Kingma, 2014) optimizer parameters $\beta_1 = 0.9$ and $\beta_2 = 0.95$. The training schedule includes a warm-up phase of 1,000 steps, followed by a learning rate decay over 10,000 steps, with a total of 20,000 training steps. The training takes around 3 days.

**Post-training configuration.** During post-training, the timestep schedule for the few-step generator is set to $\{1000, 900, 750, 500\}$. For each generator update, the fake score network is updated 4 times.

The learning rates are set to $1 \times 10^{-6}$ for the generator and $5 \times 10^{-7}$ for the discriminator, both with a weight decay of $1 \times 10^{-6}$. We use the Adam optimizer with $\beta_1 = 0.9$ and $\beta_2 = 0.95$. The training schedule consists of a 1,000 step warm-up, followed by a learning rate decay over 5,001 steps, and a total of 10,000 training steps. The GAN loss weights for both the generator and discriminator are set to $5 \times 10^{-3}$. The training takes around 2 days. The frequency of different tasks is controlled as follows: the probability ratio for training on MV-oriented mode, input views of 3D-oriented mode, and novel views of 3D-oriented mode tasks is 1:3:1. The ratio for sampling multi-view data versus out-of-distribution data is 2:1.

We use bf16 precision for both training phases. The batch size is 64, using 64 NVIDIA H20 GPUs. For distributed training, we adopt the FSDP (Fully Sharded Data Parallel) strategy and activation checkpointing to improve training efficiency and memory utilization. The prediction of MV-oriented mode is actually $v$-prediction, following the original video diffusion model (Wan et al., 2025). We use the flow matching schedule (Lipman et al., 2023) for both training phases.

**Dataset configuration.** For both pre-training and post-training, we utilize the following multi-view datasets: (1) MVImgNet (Yu et al., 2023): an object-centric dataset with a resolution of 480×704; (2) RealEstate10K (Zhou et al., 2018): an indoor scene dataset with a resolution of 704×480 and frame stride $\in [5, 6, 7, 8, 9, 10, 11, 12]$; (3) DL3DV10K (Ling et al., 2024): a general-purpose scene dataset with a resolution of 704×480 and frame stride $\in [2, 3, 4]$.

For out-of-distribution data during post-training, we employ: (1) Arbitrary image and text data paired with RealEstate10K and WorldScore camera trajectories: a general dataset with a resolution of 704×480. The images and texts are sampled from a proprietary video dataset. (2) Echo4O (Ye et al., 2025) images with WildRGBD (Xia et al., 2024) camera trajectories: a stylized, object-centric dataset with a resolution of 480×704.

## B   DETAILS OF CROSS-MODE POST-TRAINING

In Sec. 3.2, we described the specific process of Cross-mode Post-Training. Here, we provide more relevant details and pseudocodes.

**GAN loss**. The GAN loss we introduced during our cross-mode post-training is consistent with that of DMD2 (Yin et al., 2024a), which is:

$$\mathcal{L}_{\text{GAN}} = \min_{D} \max_{G_\theta} \mathbb{E}_{x,z,t} \left[ \log D(F(x,t)) + -\log \left( D(F(G_\theta(z), t)) \right) \right], \tag{7}$$

where $x$ is the real data (*i.e.*, multi-view latents $\mathcal{X}$ in our case), $F$ is the forwarding process of diffusion model, and $t$ is a random timestep. The discriminator $D$ share the same DiT network with the fake score estimator $\mu_{\text{fake}}$. Therefore, calculating the GAN loss and fake score only requires a single pass through the DiT network. When using out-of-distribution data for distillation, there is no corresponding real data. Using GAN loss in this case will lead to distribution misalignment. Therefore, we ignore GAN loss when working with OOD data during post-training.

**Few-step Generation Process**. We adapt the multi-step generation process of DMD2 (Yin et al., 2024a) to our dual-mode generator to formulate the student model. Furthermore, since 3D scene generation requires the capability of novel view synthesis, our 3D generation process also needs to support novel render cameras that are different from the input cameras. The specific algorithm is:

---

**Algorithm 1** Few-step Generation Process

---

**Input:** random noise $z \sim \mathcal{N}(0, 1)$, target timestep index $n$, conditions $y$, input cameras $\mathcal{C}$, render cameras $\mathcal{C}'$, predefined time scheduler $\alpha$ and $\sigma$, MV-oriented generator $G_{\theta,\mathrm{MV}}$, 3D-oriented generator $G_{\theta,\mathrm{3D}}$, latent encoder $E$, and generation mode.
**Output:** generated multi-view latent $G_\theta(z)$

  1:  $\mathcal{Z}_{t_1} \leftarrow z$
  2:  **for** $i = 1, 2, \ldots, n - 1$ **do**
  3:     # Reversing
  4:     $\hat{\mathcal{Z}}_{t_i} \leftarrow E(R(G_{\theta,\mathrm{3D}}(\mathcal{Z}_{t_i}, t_i, y, \mathcal{C}), \mathcal{C}))$
  5:     # Forwarding
  6:     $\epsilon \leftarrow \mathcal{N}(0, 1), \mathcal{Z}_{t_{i+1}} \leftarrow \alpha_{t_{i+1}} \hat{\mathcal{Z}}_{t_i} + \sigma_{t_{i+1}} \epsilon$
  7:  **end for**
  8:
  9:  **if** mode $=$ "MV" **then**
10:     $G_\theta(z) = \hat{\mathcal{Z}}_{t_n} \leftarrow G_{\theta,\mathrm{MV}}(\mathcal{Z}_{t_n}, t_n, y, \mathcal{C})$
11:  **else**
12:     $G_\theta(z) = \hat{\mathcal{Z}}_{t_n} \leftarrow E(R(G_{\theta,\mathrm{3D}}(\mathcal{Z}_{t_n}, t_n, y, \mathcal{C}), \mathcal{C}'))$
13:  **end if**
14:
15:  **return** $G_\theta(z)$

---

**Cross-mode Post-training Process**. Our post-training procedure aligns with that of DMD2, but is tailored to our specific data and generation pipeline. Specifically, as illustrated in Fig. 3 (right), our generator can operate in two modes: MV-oriented and 3D-oriented. When using 3D-oriented mode, we render novel views instead of using input views for loss computation with a certain probability. Meanwhile, when novel views are not used, we compute the CMC loss (Eq. 6) to stabilize the training of the 3D-oriented generator. Notably, as the MV-oriented and 3D-oriented modes share the DiT network, CMC loss computation incurs negligible additional training overhead. For data, we sample out-of-distribution data with a certain probability during training. In such cases, as no multi-view ground truth is available, we omit GAN loss computation. The detailed algorithm is:

---

**Algorithm 2** Cross-mode Post-training Process

---

**Input:** Regular multi-view data, OOD data, MV-oriented generator $G_{\theta,\text{MV}}$, 3D-oriented generator $G_{\theta,\text{3D}}$, real and fake score estimators $\mu_{\text{real}}$ and $\mu_{\text{fake}}$, discriminator $D$, latent encoder $E$

**Output:** optimized 3D-oriented generator $G_{\theta,\text{3D}}$

1: $\mu_{\text{real}} \leftarrow \mu_{\text{real}}$
2: **while** not converge **do**
3:     mode $\leftarrow$ RandomChoice(["MV", "3D"], weights $= [1, 4]$)
4:     data $\leftarrow$ RandomChoice(["Regular", "OOD"], weights $= [2, 1]$)
5:     isnovel $\leftarrow$ RandomChoice([True, False], weights $= [1, 3]$) **if** mode $=$ "3D" **else** False
6:
7:     Sample $y$, $\mathcal{C}$ and $\mathcal{C}_{\text{novel}}$ from data
8:     Sample correponding $\mathcal{X}$ and $\mathcal{X}_{\text{novel}}$ from data only if data $=$ "Regular"
9:     $\mathcal{C}' = \mathcal{C}_{\text{novel}}$ **if** isnovel **else** $\mathcal{C}$
10:    $z \leftarrow \mathcal{N}(0, 1)$
11:    $n \leftarrow$ RandomInt$(1, N)$
12:    Generate $G_\theta(z)$ with Algorithm 1
13:
14:    # Two Time-scale Update Rule in DMD2
15:    **if** iter $\% \ 5 = 0$ **then**
16:       Optimize $G_{\theta,\text{MV}}$ or $G_{\theta,\text{3D}}$ by DMD loss (Eq. 3) with $\mu_{\text{real}}$ and $\mu_{\text{fake}}$
17:       **if not** isnovel **then**
18:          Optimize $G_{\theta,\text{MV}}$ and $G_{\theta,\text{3D}}$ by CMC loss (Eq. 6)
19:       **end if**
20:       **if** data $\neq$ "OOD" **then**
21:          Optimize $G_{\theta,\text{MV}}$ or $G_{\theta,\text{3D}}$ by GAN loss (Eq. 7) with $D$
22:       **end if**
23:    **else**
24:       Optimize $\mu_{\text{fake}}$ by diffusion loss (Eq. 1)
25:       **if** data $\neq$ "OOD" **then**
26:          Optimize $D$ by GAN loss (Eq. 7)
27:       **end if**
28:    **end if**
29:
30:    iter $\leftarrow$ iter $+ 1$
31: **end while**
32: **return** $G_{\theta,\text{3D}}$

---

## C  RELATED WORKS

**Iterative 3D scene generation.** Recent advances in diffusion models (Rombach et al., 2022; Podell et al., 2023; Zhang et al., 2023) have enabled iterative generation of 3D scenes. DiffDreamer (Cai et al., 2023) improves multi-view consistency by conditioning on both past and future frames. SceneScape (Fridman et al., 2023), Text2Room (Höllein et al., 2023), and RGBD2 (Lei et al., 2023) refine mesh-based representations through depth-conditioned diffusion. WonderJourney (Yu et al., 2024a) leverages point clouds with VLM-guided re-generation. Text2NeRF (Zhang et al., 2024a) and 3D-SceneDreamer (Zhang et al., 2024b) address error accumulation by utilizing NeRF (Mildenhall et al., 2020) representations. LucidDreamer (Chung et al., 2023), WonderWorld (Yu et al., 2025), RealmDreamer (Shriram et al., 2025), and WonderTurbo (Ni et al., 2025) accelerate generation and enhance fidelity using 3DGS (Kerbl et al., 2023). While iterative generation methods have made significant progress, they often suffer from cross-view semantic inconsistency. In contrast, data-driven approaches leverage rich cross-view priors to better maintain semantic coherence.

**Multi-view-oriented 3D scene generation.** A major class of data-driven methods adopts a two-stage pipeline: generate multi-view images first, then reconstruct. CAT3D (Gao et al., 2024) synthesizes novel views via multi-view diffusion, followed by 3D reconstruction. DimensionX (Sun et al., 2024) generates temporally coherent videos, expands viewpoints through video diffusion, and reconstructs 3D scenes from frames. ODIN (Wallingford et al., 2024) produces trajectory-conditioned novel views

for subsequent reconstruction. GenXD (Zhao et al., 2025) decouples multi-view and temporal features to jointly generate static and dynamic scenes. Bolt3D (Szymanowicz et al., 2025) outputs colored 3D Gaussians from images and point maps generated by multi-view diffusion. Prometheus (Yang et al., 2025) leverages the training paradigm of RGBD latent diffusion models. SplatFlow (Go et al., 2025a) jointly learns camera poses and multi-view image distributions from text. Wonderland (Liang et al., 2025) generates continues multi-view latents via video diffusion, then reconstructs scenes using latent-based reconstruction models. AniGS (Qiu et al., 2025) generates multi-view RGB and normal images and propose a strategy to reconstruct 4DGS with inconsistent views.

**3D-oriented 3D scene generation.** Another line of work adopts a 3D-oriented pipeline, employing rendering during denoising steps. DMV3D (Xu et al., 2023) introduces a large reconstruction-based denoising model based on a triplane NeRF representation, performing denoising through NeRF-based reconstruction and rendering. Dual3D (Li et al., 2024a) proposes a dual-mode multi-view latent diffusion model based on pre-trained image diffusion models and neural surface rendering to reduce training and rendering costs. VideoMV (Zuo et al., 2024) proposes a 3D-aware sampling strategy to enhance multi-view consistency during denoising. Director3D (Li et al., 2024b) synthesizes pixel-aligned 3D Gaussians directly from latent space using trajectory-conditioned multi-view diffusion, followed by SDS++ refinement. DiffusionGS (Cai et al., 2024) presents a diffusion model that outputs pixel-aligned 3DGS at each timestep to ensure 3D consistency. Cycle3D (Tang et al., 2025) proposes a unified generation-reconstruction framework, where the 3D reconstruction module is integrated into the multi-step denoising process to further guarantee 3D consistency.

**Distillation for diffusion models.** Distillation techniques for diffusion models focus on transferring knowledge from a pretrained teacher model to a more compact and efficient student model. Denoising Student (Luhman & Luhman, 2021) achieves this by training a single-step generator to minimize the RMSE between the outputs of the teacher and student models. Consistency Model (Song et al., 2023) enables trajectory distillation, allowing the student to mimic the teacher's denoising process across multiple steps. Adversarial Diffusion Distillation (ADD) (Sauer et al., 2024b), Latent Adversarial Diffusion Distillation (LADD) (Sauer et al., 2024a), Adversarial Post-Training (APT) (Lin et al., 2025a), and Autoregressive Adversarial Post-Training (AAPT) (Lin et al., 2025b) further enhance distillation by introducing adversarial objectives to improve the student performance. Distribution Matching Distillation (DMD) (Yin et al., 2024b) formulates the distillation objective as optimizing the reverse KL-divergence between the student and teacher distributions. DMD2 (Yin et al., 2024a) extends this framework by incorporating a GAN-based objective and supporting for multi-step generators, further improving the flexibility and effectiveness of the distillation process.

## D  LLM USAGE CLARIFICATION

Large Language Models (LLMs) were utilized solely for language refinement in the preparation of this manuscript. All LLM-generated content has been thoroughly reviewed and validated by the authors, who assume full responsibility for the entirety of the paper.

## E  LIMITATIONS

While the proposed `FlashWorld` demonstrates strong capabilities in generating high-fidelity and efficient 3D scenes, several limitations remain. First, despite increasing the number of views, the diversity and scale of generated scenes are still constrained by the coverage of existing datasets. Second, the model currently struggles with accurately generating fine-grained geometry, mirror reflections, and articulated objects. These issues may be alleviated by incorporating depth priors (Yang et al., 2024a;b; Chen et al., 2025) and more 3D-aware structural information (Jiang et al., 2025; Wang et al., 2025) to further enhance the quality of our pixel-aligned 3D Gaussians.

## F  RGBD RENDERING RESULTS

While `FlashWorld` does not explicitly incorporate depth supervision, the 3DGS outputs inherently enable the export of depth maps. In this regard, we present several RGBD rendering results in Fig. 9. This serves to demonstrate that our model is capable of learning meaningful depth geometric information solely via image supervision.

In Fig. 10, we show a comparison of the depth rendered by our method with that of Director3D and Prometheus. It can be seen that our model not only produces clearer and more realistic RGB rendering results, but also achieves more accurate depth results.

## G  COMPARISON TO CAMERA-CONTROLLABLE VIDEO GENERATION MODELS

In the main paper, all core compared baselines are capable of generating 3D representations. In Fig. 11, we present a visual comparison of novel view synthesis results between our method and ViewCrafter (Yu et al., 2024b), a camera-controllable video generation model. It can be observed that the 3DGS rendering results of `FlashWorld` are comparable in visual quality to the video generation results of ViewCrafter. Additionally, `FlashWorld` achieves better color preservation of the input image (*e.g.*, the identity of the cat in the left column) and outperforms in certain details (*e.g.*, the bushes in the right column). Notably, on one A100 GPU, ViewCrafter requires 2 minutes for generation, while `FlashWorld` only takes 7 seconds. After generating the scene, `FlashWorld` can perform rendering at real time, while ViewCrafter needs to re-run the diffusion.

## H  HANDLING OCCLUDED AND TRANSPARENT OBJECTS

For occlusions, `FlashWorld` can automatically infer and complete occluded objects by designing appropriate camera positions. We present an example in Fig. 12 (bottom) where `FlashWorld` handles a scenario where most of a car is occluded, yet completes it via view synthesis. For transparent objects, since 3DGS employs volume rendering, transparent objects can generally be modeled using low opacity values. We show an example in Fig. 12 (top) where `FlashWorld` handles a scene inside a semi-transparent fish tank.

## I  MORE RESULTS

We provide more generation results in Fig. 13, including object-centric, indoor, outdoor, realistic, and stylized scenes, to demonstrate the strong and generalizable generation ability of our model.

For video rendering results, please kindly refer to this anonymous website: `https://anonymous.4open.science/w/FlashWorld_Page-5FAD/`.

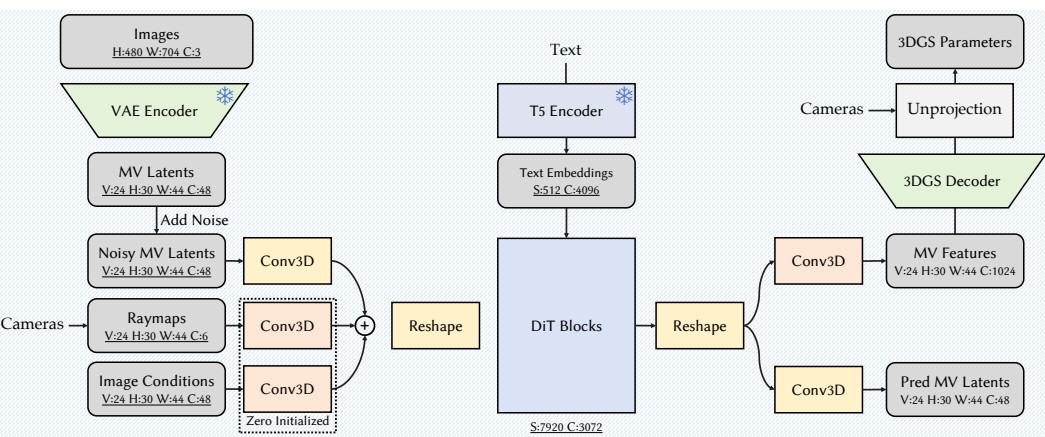

Figure 8: **Architecture of the dual-mode multi-view latent diffusion model.**

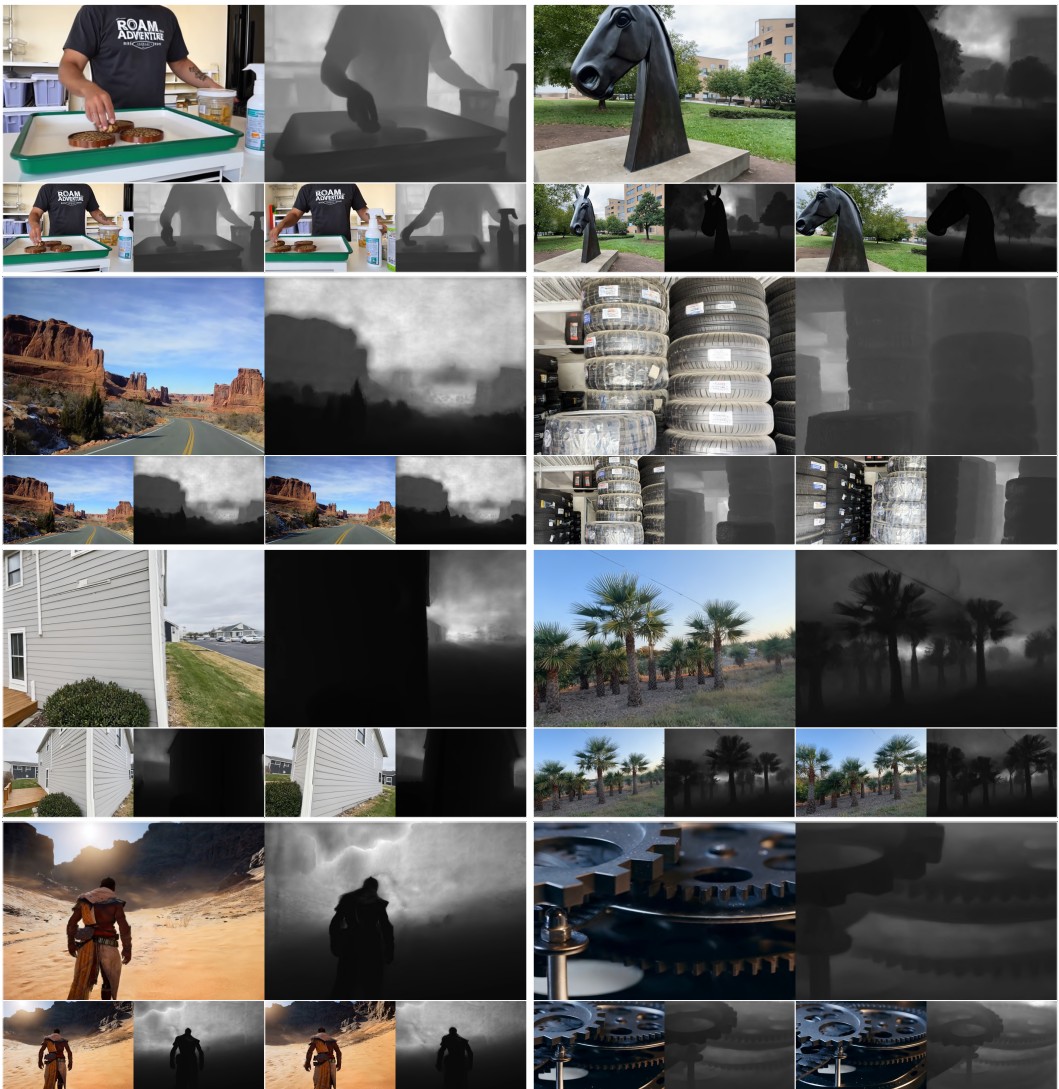

Figure 9: **RGBD rendering results.**

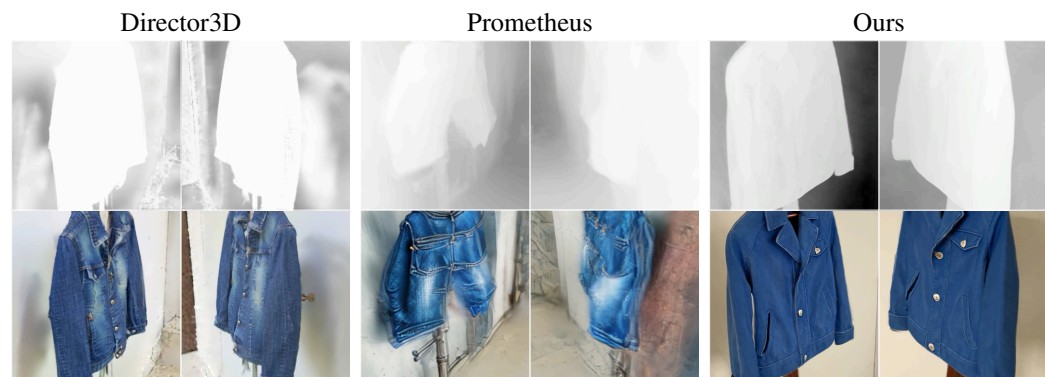

Director3D          Prometheus          Ours

Figure 10: **RGBD rendering results of different methods.**

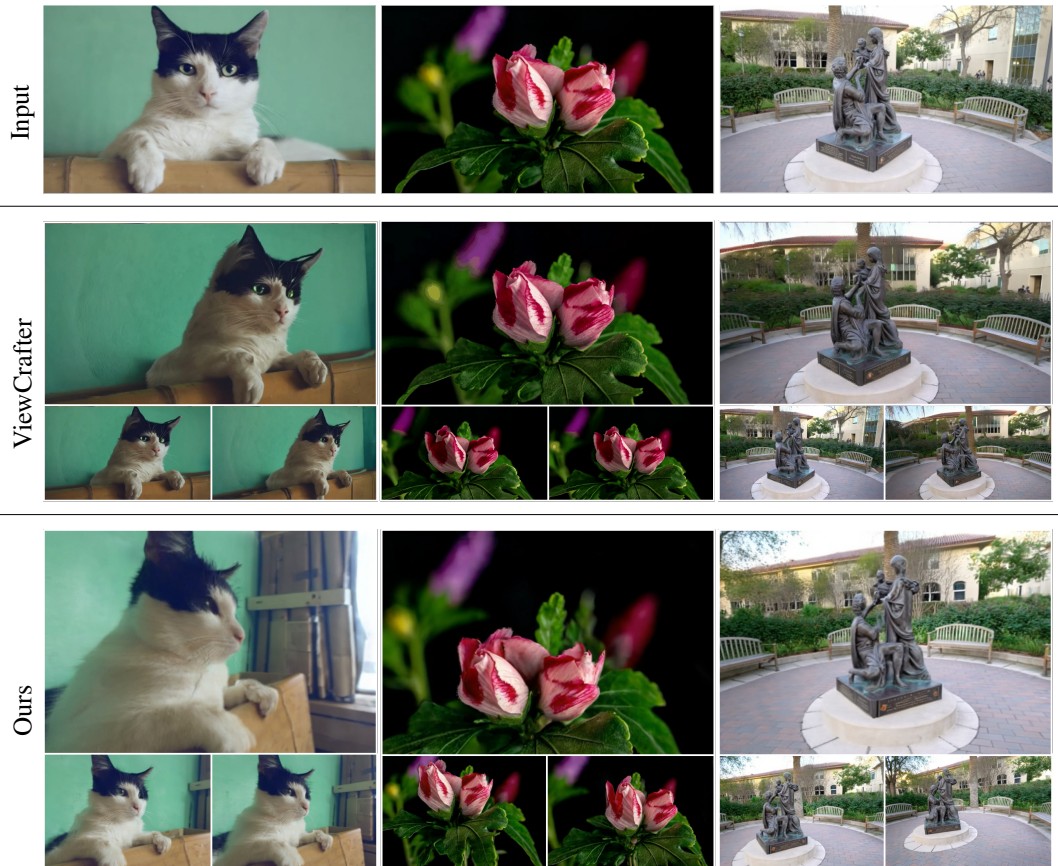

Figure 11: **Novel view synthesis results compared to ViewCrafter (Yu et al., 2024b).**

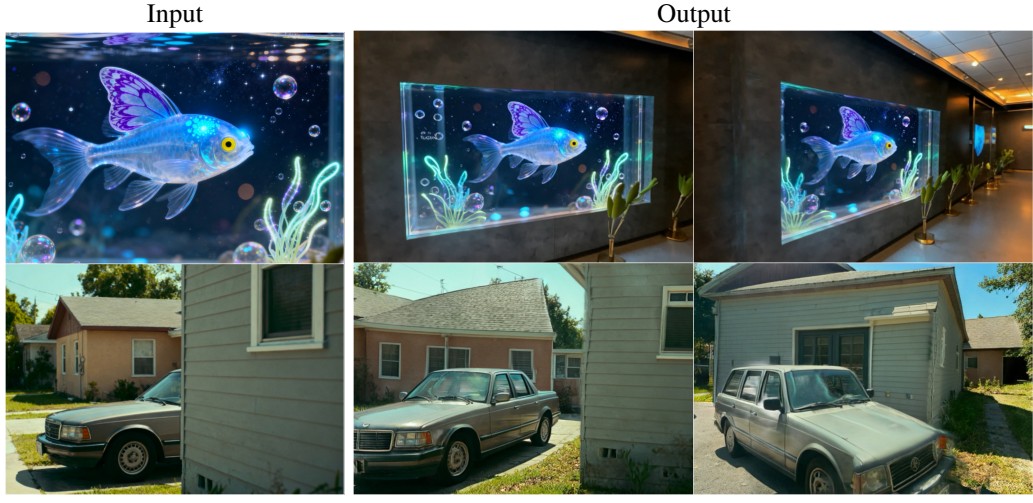

Figure 12: **FlashWorld is capable of handling occluded and (semi-)transparent objects.**

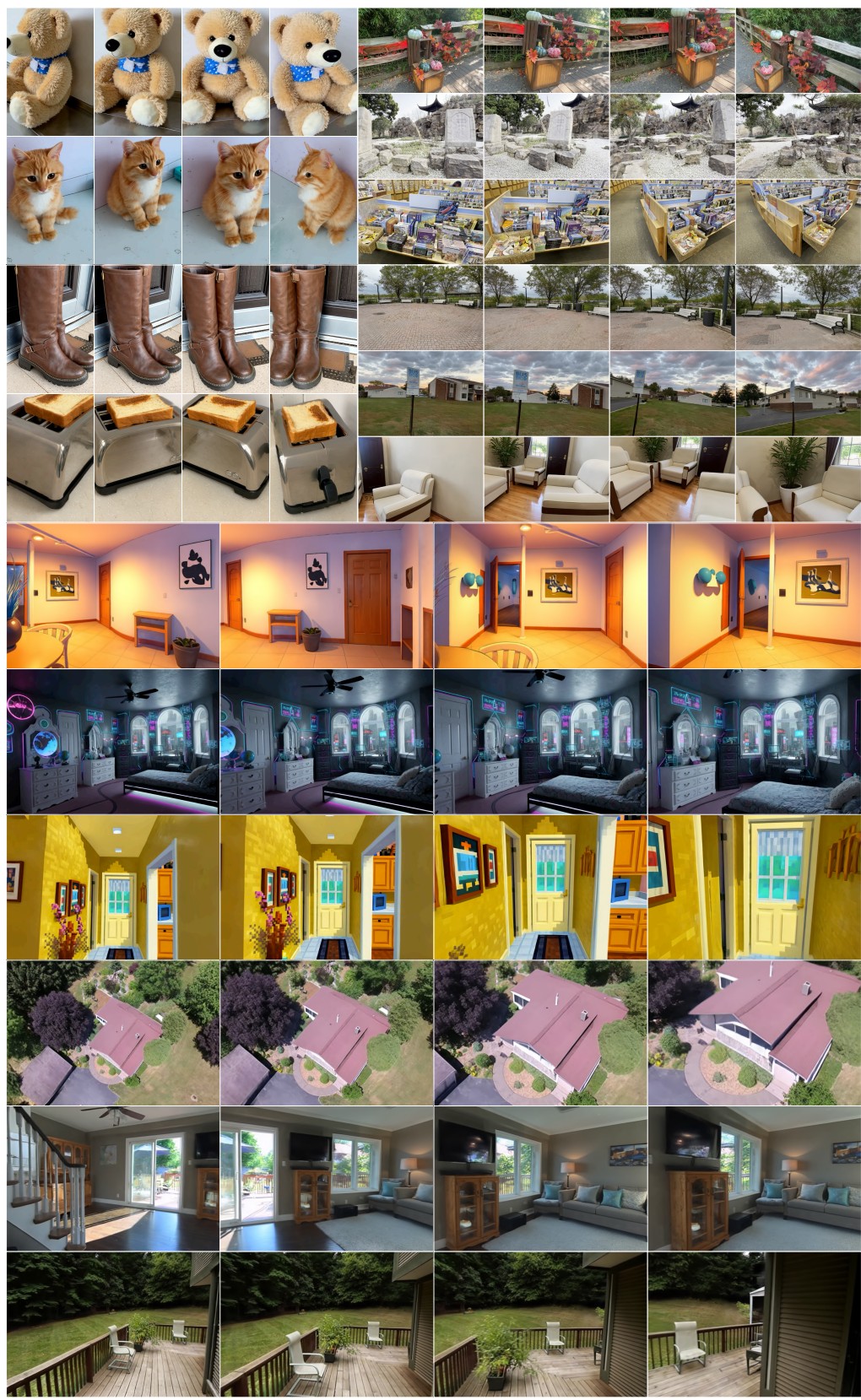

Figure 13: **More generation results.** All images are rendered with generated 3DGS.