# OpenReview forum: "FlashWorld: High-quality 3D Scene Generation within Seconds"
_ICLR.cc/2026/Conference — ICLR 2026 Oral_

### Official Review · Reviewer_mRkm · 2025-10-27

**Soundness:** 3
**Presentation:** 2
**Contribution:** 3
**Rating:** 6
**Confidence:** 4

**Summary:**

The paper addresses text- and image-to-scene generation. Previous methods that leverage powerful video foundation models (VDMs) face a trade-off: multi-view-oriented approaches lack 3D consistency, while 3D-oriented methods often yield poor visual quality. This paper introduces pre-training and post-training strategies for VDMs to improve 3D consistency and accelerate video generation. For pre-training, the authors add a 3DGS decoder (the "3D-oriented" branch) to inject 3D priors and enhance the latent video diffusion model’s 3D consistency. For post-training, they distill a multi-view-oriented teacher into a 3D-oriented student to speed up generation. Comprehensive experiments show the proposed methods outperform prior state-of-the-art approaches in both qualitative and quantitative evaluations.

**Strengths:**

- The two proposed training strategies are novel, efficient, and effective for 3D scene generation.

- The authors present comprehensive experiments that convincingly support their claims.

- Comparison results show the proposed method outperforms both image-to-3D and text-to-3D approaches, producing finer-detail renderings and faster generation times.

**Weaknesses:**

- The writing flow is poor and Sec. 3.2 feels chaotic. For example, the sentence To generate a 3D scene, the 3D-oriented multi-view generation process alternates between denoising and noise injection steps to enhance sample quality.'' is confusing: is the goal to generate a 3D scene, to enhance sample quality, or both?

- For Sec. 3.3, the reason the model improves the quality of out-of-domain (OOD) data is unclear. As I understand it, the approach uses camera-trajectory augmentation to enhance the model’s generalizability and also discards the GAN loss during training. Which of these two changes is more important for OOD performance? Data augmentation is easy to see as a way to improve generalization, but why does including a GAN loss lead to poor generalizability?

**Questions:**

1. There may be a typo on lines 254--255. You refer to λ in relation to Eq.(6), but Eq.(6) does not contain λ.

2. Missing some relative references:
- VideoMV: Consistent Multi-View Generation Based on Large Video Generative Model,  which proposes a 3D-aware sampling strategy;
- AniGS: Animatable Gaussian Avatar from a Single Image with Inconsistent Gaussian Reconstruction, using 4DGS to alleviate the inconsistency from multi-view video generation.

Please consider add the above relative references.

In summary, the authors propose interesting pre-training and post-training strategies for video-diffusion model training to improve 3D consistency and quality in scene generation, achieving state-of-the-art results in both quantitative and qualitative experiments. However, the writing flow is poor: some sentences are confusing, and I have concerns about the model’s generalizability. As a result, I am inclined to give a borderline-accept score, though I would be happy to raise it if the authors address these main concerns.

---

> ### Author Response · Authors · 2025-11-14
>
> ***To Reviewer #mRkm***:
>
> We thank the reviewer for your appreciation of the **novel, efficient and effective strategies**, and the **comprehensive experiments** of FlashWorld. We address your weaknesses and questions point-by-point below.
>
> ---
>
> ***W1: Writing of Sec. 3.2.***
>
> We sincerely appreciate the reviewer’s feedback. To clarify the detailed procedure of FlashWorld post-training, we have added comprehensive descriptions of the Few-step Generation Process and Cross-mode Post-training Process, along with corresponding pseudocode in **[Appendix B, Lines 816-898]**. Regarding the sentence you mentioned, we have revised it in **[Lines 232-233]** to improve clarity and logical coherence.
>
> ***W2: About OOD Co-training and GAN loss.***
>
> We’d love to clarify that our proposed OOD co-training is not limited to camera trajectory augmentation, but rather enhances **all conditions** (including input images, texts, and trajectories). The construction details can be found in **[Appendix A, Lines 810-814]**. This design inherently leverages the property of DMD loss (Eq. 2) that it does not require multi-view ground truth, along with the strong generalizability of the MV-oriented mode which is also trained on video data, to improve the 3D-oriented mode’s generalization.
>
> Introducing GAN loss does **NOT** inherently causes poor generalization. However, when distilling with OOD data, there is no corresponding multi-view ground truth. In this case, using GAN loss would lead to misalignment between real and fake distributions, making the discriminator easily distinguish fake samples. Thus, we only apply GAN loss when using regular multi-view data. This is a necessary strategy for OOD data co-training. You can refer to **[Appendix B, Lines 881-893]** for the algorithm details.
>
> To further validate the effect of leveraging OOD data for post-training, we conducted quantitative evaluations on the w/o OOD model using 2000 samples from the WorldScore Benchmark. As observed, while the w/o OOD model achieves slight improvements in certain metrics compared to our full model, it exhibits notable regressions in object control, content alignment, and subjective quality. Additionally, the average score of the w/o OOD model is substantially lower.
>
> |  | 3d_consistency | photometric_consistency | object_control | content_alignment | style_consistency | subjective_quality | Average |
> | --- | --- | --- | --- | --- | --- | --- | --- |
> | w/o OOD | **87.35** | **90.12** | 34.94 | 44.87 | **83.16** | 46.24 | 64.45 |
> | Full model | 85.87 | 86.72 | **49.61** | **53.96** | 81.52 | **54.63** | **68.72** |
>
> ---
>
> ***Q1: Missing $\lambda$ in Eq. 6.***
>
> We sincerely apologize for this typo. We have added the parameter to **[Eq. 6, Lines 252-254]** in the revision.
>
> ***Q2: Missing References.***
>
> We sincerely appreciate your valuable suggestion. We have added citations to these works in **[Lines 095-096]**, **[Lines 925-926]** and **[Lines 931-933]**.
>
> ---
>
> We welcome any further comments, questions, and discussions from you.

---

> ### Comment · Reviewer_mRkm · 2025-11-24
> **Official Comment by  Reviewer mRkm**
>
> I appreciate the feedback provided by authors. I have no further questions and I vote for acceptance.
>
> ---
> By the way, the reference for AniGS you cite is wrong, please check it again.

---

> > ### Author Response · Authors · 2025-11-24
> >
> > We sincerely appreciate you for your positive voting. Your suggestions greatly helped improve the comprehensiveness of our paper.
> >
> > We fix the reference for AniGS in our latest revision **[Lines 930~932]**.

---

### Official Review · Reviewer_m4Bg · 2025-11-01

**Soundness:** 3
**Presentation:** 3
**Contribution:** 3
**Rating:** 6
**Confidence:** 4

**Summary:**

The paper tackles scene generation from text and images. Existing video foundation model approaches trade off between multi-view methods, which struggle with 3D consistency, and 3D-focused methods, which often compromise visual fidelity. To overcome this, the authors propose pre-training and post-training techniques for VDMs: a 3DGS decoder (the “3D-oriented” branch) is added during pre-training to inject geometric priors and improve 3D coherence, and a teacher–student distillation step transfers knowledge from a multi-view-oriented teacher to a 3D-oriented student to speed up generation.

**Strengths:**

- The paper is clearly written and easy to follow.

- The organization is logical, and most design choices are validated with ablation studies.

- The method shows notable improvements over prior work, particularly in preserving fine-grained details.

**Weaknesses:**

The paper is novel and effective; the following points are offered as constructive suggestions for further improvement.

- The diversity and scale of generated scenes are still constrained by the coverage of existing datasets.

- The model currently struggles with accurately generating fine-grained geometry, mirror reflections, and articulated objects. These issues may be alleviated by incorporating depth priors and more 3D-aware structural information

- Although FlashWorld does not use explicit depth supervision, its 3D Gaussian Splatting (3DGS) outputs can be used to extract depth maps. However, the quality of the resulting depth information could be improved.

**Questions:**

There may be an error in Eq. (6), where is \lambda ?

**Details Of Ethics Concerns:**

There are no obvious ethics issues.

---

> ### Author Response · Authors · 2025-11-14
>
> ***To Reviewer #m4Bg***:
>
> We thank the reviewer for your appreciation of the **novel and effective paper**, the **logical organization ,** and the **notable improvements over prior works** of FlashWorld. We address your weaknesses and questions point-by-point below.
>
> ---
>
> ***W1: The diversity and scale are constrained by existing datasets.***
>
> We would like to highlight that FlashWorld has incorporated multiple strategies to mitigate this limitation as much as possible: leveraging pre-trained video diffusion models to harness prior knowledge, performing joint training on video data during pre-training, and proposing out-of-distribution co-training to expand the acceptable distribution of the 3D-oriented generator using general text and image data. We emphasize that our method has achieved SOTA results under the constraints of existing datasets with extensive experiments.
>
> ***W2: The model struggles with fine-grained geometry, mirror refections, and articulated objects.***
>
> We would like to clarify that these limitations primarily stem from the inherent constraints of 3DGS representations, rendering operators, and video foundation models, rather than the proposed methods in this paper. As rendering schemes and video foundation models continue to advance in the future, along with targeted improvements for these specific issues, our proposed dual-mode pre-training, cross-mode post-training, and OOD co-training strategies are general and compatible with different 3D representations, rendering operators, and video foundation models.
>
> ***W3: The depth quality can be improved.***
>
> We would like to clarify that the depth results presented in the Appendix are entirely derived from multi-view RGB supervision. We have also added some rendered depth comparisons between our method and several open-source baselines for the text-to-3D scene generation task in **[Fig. 9, Lines 1065-1079]**. We acknowledge that the depth results are not yet perfect and have room for improvement. However, we have also attempted to regularize the rendered depth using monocular depth estimation methods such as DepthAnything. We found that, given the high generation fidelity of our current model, coarse monocular depth can not effectively enhance the quality of the rendered depth. In this paper, to focus on our core contributions, we did not further improve the depth regularization term. However, we will prioritize this as an important future research direction.
>
> ---
>
> ***Q1: Missing $\lambda$ in Eq. 6.***
>
> We sincerely apologize for this typo. We have added the parameter to **[Eq. 6, Lines 252-254]** in the revision.
>
> ---
>
> We welcome any further comments, questions, and discussions from you.

---

> > ### Comment · Reviewer_m4Bg · 2025-11-24
> > **Official Comment by Reviewer m4Bg**
> >
> > I read the other reviewers' comments and the authors' response. Thanks to the authors for their feedback
> >  my most concerns have been addressed, and I will maintain my recommendation for acceptance.

---

> ### Author Response · Authors · 2025-11-24
>
> We sincerely appreciate your supportive vote and constructive suggestions on FlashWorld. Since a score of 6 remains borderline for acceptance, we would like to briefly emphasize our work's core value, in the hope that you may reconsider your rating with a stronger recommendation:
>
> *FlashWorld’s core insight lies in enabling a theoretically 3D-consistent generator to improve visual quality by distilling a multi-view (or video) generator. It simultaneously guarantees three key advantages in 3D scene generation—quality, generation speed, and 3D consistency—with reasonable training costs and minimal architecture modifications. This makes FlashWorld currently a method with huge inference speed advantages while maintaining SOTA visual quality, even against concurrent works.*
>
> Thank you again for your time. In any case, we sincerely appreciate your efforts and fully respect any decision you reach.

---

### Official Review · Reviewer_N491 · 2025-11-01

**Soundness:** 3
**Presentation:** 2
**Contribution:** 3
**Rating:** 6
**Confidence:** 4

**Summary:**

This work presents a generative model that produces high-quality 3D scenes from a single image or text prompt in just seconds—10-100× faster than prior methods. FlashWorld’s key innovation is a distillation strategy that transfers high visual fidelity from a multi-view-oriented diffusion teacher to a 3D-oriented student, improving 3D consistency. The authors also introduce an out-of-distribution co-training strategy to improve generalization to scenes beyond the training distribution. Extensive experiments show FlashWorld outperforms state-of-the-art methods in both generation quality and inference speed.

**Strengths:**

- The paper is clearly written and easy to follow.
- Experimental validation is thorough, with a comprehensive benchmark against many competing methods.
- The out-of-distribution co-training approach is an effective and efficient way to improve generalization and robustness.

**Weaknesses:**

-  Although the model demonstrates strong and generalizable generation capabilities, its video-rendering performance is not extensively evaluated in the main paper.
- As noted in the limitations, the model still struggles with fine-grained geometry, mirror reflections, and articulated objects.

**Questions:**

see weakness

---

> ### Author Response · Authors · 2025-11-14
>
> ***To Reviewer #N491***:
>
> We thank the reviewer for your appreciation of the **clearly written paper**, the **thorough experimental validation,** and the **effective and efficient way to improve generalization and robustness** of FlashWorld. We address your weaknesses and questions point-by-point below.
>
> ---
>
> ***W1: Video rendering performance should be evaluated.***
>
> We sincerely appreciate the reviewer’s valuable suggestion. We would like to clarify that the metrics in Tab. 1 (Q-Align IQA, Q-Align IAA, CLIP IQA+, CLIP Aesthetic) and Tab. 2 (Subjective Quality) are evaluated based on frames from the rendered videos. To further complement the video rendering performance assessment, we have additionally evaluated an extra metric: Q-align VQA, a state-of-the-art no-reference video quality assessment metric based on VLLMs. The results are presented as follows:
>
> **Table: Q-align VQA scores of rendered videos of different methods (higher is better)**
>
> |  | T3Bench-200 | DL3DV-200 | WorldScore-200 |
> | --- | --- | --- | --- |
> | Director3D | 0.4841 | 0.6682 | 0.5388 |
> | Promtheus | 0.4168 | 0.5088 | 0.5839 |
> | Ours | **0.7676** | **0.8286** | **0.7219** |
>
> As shown in the table, FlashWorld consistently outperforms the baseline methods by a significant margin in video quality assessment, further validating the effectiveness of our proposed framework for video rendering.
>
> ***W2: The model struggles with fine-grained geometry, mirror refections, and articulated objects.***
>
> We would like to clarify that these limitations primarily stem from the inherent constraints of 3DGS representations, rendering operators, and video foundation models, rather than the proposed methods in this paper. As rendering schemes and video foundation models continue to advance in the future, along with targeted improvements for these specific issues, our proposed dual-mode pre-training, cross-mode post-training, and OOD co-training strategies are general and compatible with different 3D representations, rendering operators, and video foundation models.
>
> ---
>
> We welcome any further comments, questions, and discussions from you.

---

> > ### Author Response · Authors · 2025-11-26
> >
> > Dear Reviewer,
> >
> > we sincerely appreciate your positive voting and valuable suggestions. Since there are only a few days left in the discussion period, to ensure that we have sufficient time to provide additional materials later, we kindly ask if our responses have addressed your concerns.
> >
> > We also welcome any further questions you may have and will actively respond to them.
> >
> > Best regards,
> >
> > FlashWorld Authors

---

> > > ### Comment · Reviewer_N491 · 2025-11-26
> > >
> > > Thanks for the response of authors. And after checking the comments of other reviewers and the feedback, I have no further issues.

---

> > > > ### Author Response · Authors · 2025-11-26
> > > >
> > > > We are glad that our responses have addressed your concerns. We would like to kindly request that you reconsider the rating for a stronger recommendation. Your support would greatly encourage us. In any case, we sincerely appreciate your time and respect your decision.

---

### Official Review · Reviewer_wnKK · 2025-11-01

**Soundness:** 3
**Presentation:** 3
**Contribution:** 3
**Rating:** 6
**Confidence:** 4

**Summary:**

This paper introduces FlashWorld, a framework for fast and high-quality 3D scene generation that addresses the critical speed-quality trade-off in current methods. The authors propose a cross-mode distillation approach that leverages both MV-oriented (multi-view) and 3D-oriented generation modes. The key innovation lies in using dual-mode pre-training followed by cross-mode post-training, where the MV-oriented mode serves as a teacher to provide visual quality while the 3D-oriented mode acts as a student to ensure geometric consistency.

**Strengths:**

- The idea of training a dual-mode model and using cross-mode distillation for post-training is creative and well-motivated. Using the MV-oriented mode as teacher for visual quality while training the 3D-oriented mode for consistency is an elegant solution to the speed-quality trade-off.

- Achieving ~9 second generation time while maintaining SOTA quality represents a significant practical contribution. The 10-100× speedup over prior work (CAT3D, Wonderland, etc.) makes this approach much more suitable for real-world applications.

**Weaknesses:**

- Section 3.2 (cross-mode post-training) needs more detail. This stage appears to integrate DMD2 with the dual-mode diffusion model, but the step-by-step procedure is unclear. Please add a training algorithm box or pseudocode to clarify the process.
- Include NVS comparisons with video-based methods such as TrajectoryCrafter, GEN3C, and ViewCrafter.
- Provide depth visualizations of generated scenes and compare against baselines.

**Questions:**

- Does the Out-of-Distribution (OOD) data improve only text-to-3D, or does it also help image-to-3D? The gains for text-to-3D are clear, but the impact on image-to-3D is not.
- Line 249 is confusing: “we additionally update an MV-oriented student model at a lower frequency.” From Fig. 3, it seems the 3D- and MV-oriented modes share the same DiT backbone, whose latent output feeds the 3DGS decoder. If you update the 3D-oriented model, does that also update the DiT (and therefore the MV-oriented model)? Please clarify which parameters are shared or frozen during training and how updates are applied.
- How does the method handle scenes with significant occlusions or transparent objects?

---

> ### Author Response · Authors · 2025-11-14
>
> ***To Reviewer #wnKK***:
>
> We thank the reviewer for your appreciation of the **creative and well-motivated idea**, the **elegant solution,** the **SOTA quality**, and the **significant practical contribution** of FlashWorld. We address your weaknesses and questions point-by-point below.
>
> ---
>
> ***W1: Sec.3.2 needs more details.***
>
> We sincerely appreciate the reviewer’s feedback. To clarify the detailed procedure of our post-training, we have added comprehensive descriptions of the Few-step Generation Process and Cross-mode Post-training Process, along with corresponding pseudocode in **[Appendix B, Lines 816-898]**.
>
> ***W2: Comparison with camera-controllable video generation models.***
>
> We would like to clarify that the first stage of Wonderland and VideoRFSplat (two of our baselines) are camera-controllable video generation models. As we aim to emphasize that the task is 3D scene generation, we compare the rendered results of 3DGS for all methods. Generally speaking, the video results of such methods are clearer than the rendered results of 3DGS, but suffer from camera inconsistency and dynamic artifacts.
>
> To further compare the methods you mentioned, we present novel view synthesis results of our method and ViewCrafter in **[Fig. 10, Lines 1080-1010]**. It can be observed that the 3DGS rendering results of FlashWorld are comparable in visual quality to the video generation results of ViewCrafter. This validates the effectiveness of our distillation method. Additionally, FlashWorld achieves better preservation of the input image (e.g., the identity of the cat in the left column) and outperforms in certain details (e.g., the bushes in the right column).
>
> Also, on a single A100 GPU, ViewCrafter requires 2 minutes for generation, while FlashWorld only takes 7 seconds. Furthermore, after generating the scene, FlashWorld can perform novel view rendering at real-time speed, whereas ViewCrafter needs to re-run the diffusion process.
>
> ***W3: Depth visualization and comparison.***
>
> We sincerely appreciate this valuable suggestion. You can refer to **[Fig. 8, Lines 1026-1062]** for our depth visualization. We have also added some rendered depth comparisons between our method and several open-source baselines for the text-to-3D scene generation task in **[Fig. 9, Lines 1065-1079]**. Although we do not incorporate explicit depth supervision, our depth quality is still superior to that of other baselines.
>
> ---
>
> ***Q1: Does the OOD data also improve image-to-3D generation?***
>
> Yes, OOD data also expands the acceptable distribution of input images for our model. To further validate this, we conducted a quantitative evaluation on the w/o OOD model using 2000 samples from the WorldScore Benchmark. As observed, while the w/o OOD model achieves slight improvements in certain metrics compared to our full model, it exhibits notable degradation in object control, content alignment, and subjective quality. Additionally, the average score of the w/o OOD model is substantially lower.
>
> |  | 3DConsistency | PhotometricConsistency | ObjectControl | ContentAlignment | StyleConsistency | SubjectiveQuality | Average |
> | --- | --- | --- | --- | --- | --- | --- | --- |
> | w/o OOD | **87.35** | **90.12** | 34.94 | 44.87 | **83.16** | 46.24 | 64.45 |
> | Full model | 85.87 | 86.72 | **49.61** | **53.96** | 81.52 | **54.63** | **68.72** |
>
> ***Q2: Post-training Details.***
>
> First, we refer you to the detailed algorithm pseudocode added in our response to W1, which elaborates on the schemes and update rules for using different modes and data during training. Below are specific answers to your sub-questions:
>
> ***Whether updating the 3D-oriented model also updates the DiT?*** Yes, gradients are backpropagated to optimize the DiT. However, this does not directly improve the prediction of the MV-oriented generator. In practice, the features fed into the 3DGS Decoder are not the direct prediction of the MV-oriented generator, but rather the feature maps from a separate head.
>
> ***Regarding shared or frozen parameters during training.*** Only the real score estimator is frozen, while all other modules are trainable. The MV-oriented and 3D-oriented generators share a DiT network, while the discriminator and the fake score estimator share another.
>
> ***Q3: How does FlashWorld handle occlusions and transparent objects?***
>
> FlashWorld can automatically infer and complete occluded objects by designing appropriate camera poses.
> We present an example in **[Fig. 11 (bottom), Lines 1123-1132]** where FlashWorld handles a scenario where most of a car is occluded, yet completes it via view synthesis.
> For transparent objects, since 3DGS employs volume rendering, transparent objects can generally be modeled using lower opacity values. We show an example in **[Fig. 11 (top), Lines 1115-1123]** where FlashWorld handles a scene containing a semi-transparent fish tank.
>
> ---
>
> We welcome any further comments, questions, and discussions from you.

---

> > ### Comment · Reviewer_wnKK · 2025-11-24
> >
> > I thank the authors for the detailed response. I really appreciate the effort, and most of my concerns have been resolved. However, I am still having trouble understanding the architecture behind the dual-mode diffusion model.
> >
> > The response and the newly added pseudocode emphasize that the MV mode and 3D mode models can be optimized separately, which implies they would have different parameters other than the DiT backbone. However, based on the paper, I assumed that the 3D mode simply adds a decoder on top of the MV mode (which uses the DiT backbone to produce multiview image latents). Therefore, if the MV mode model is optimized, shouldn't the 3D mode also change, even if you are not optimizing for the 3D mode?
> >
> > There is only one place that provides details about this (Lines 197–199):
> > > The denoising network is a Diffusion Transformer (DiT) enhanced with 3D attention blocks, and outputs both a denoised estimate Z_MV and an auxiliary multi-view feature F
> >
> > I am not sure if the MV output is produced by an extra MV head on top of the DiT backbone, or if the DiT directly outputs the multi-view image latent. Also, where does this auxiliary multi-view feature come from? Is it just directly using the image latent?

---

> ### Author Response · Authors · 2025-11-25
>
> We sincerely appreciate your thorough reading and interest in FlashWorld.
>
> To comprehensively and clearly illustrate the architecture of our dual-mode multi-view latent diffusion model, we added a network architecture diagram in the revision **[Fig. 8, Lines 1011~1025]**.
>
> Below are the specific responses to your questions:
>
> ---
>
> ***Q1: If the MV mode model is optimized, shouldn't the 3D mode also change, even if you are not optimizing for the 3D mode?***
>
> Yes, your understanding is correct. The DiT network is shared between the two modes and thus optimized for both.
> During the pre-training phase, the losses of the two modes are optimized simultaneously. During the post-training phase, due to GPU memory constraints, they are optimized separately. The 3D mode is trained more frequently in this stage, since the MV mode only serves as a regularization term.
>
> ***Q2: Whether the MV output is produced by an extra MV head on top of the DiT backbone, or if the DiT directly outputs the multi-view image latent?***
>
> This depends on how one interprets the structure of the DiT network. Since attention mechanisms inherently require flattening features into 1D sequences, the input and output layers of the DiT inherently include corresponding (un)patchify and reshape operations. Additionally, a Linear or Conv head is consistently integrated to adjust the channel dimension before or after the (un)patchify operations, as needed.
>
> ***Q3: Where does this auxiliary multi-view feature come from? Is it just directly using the image latent?***
>
> As illustrated in the figure, the auxiliary multi-view feature is processed by a separate head, and the image latent cannot be used directly. The reason is that our 3DGS decoder does not inherently possess the capability to model interactions between multi-view features. Instead, it only performs decoding and upsampling of multi-view features (with already embedded 3D information of the auxiliary multi-view feature) into 3DGS parameters. All interactions between multi-view features as well as the extraction of 3D information are comprehensively handled by the DiT itself. Thus, unlike LaLRM proposed in Wonderland [1], we do not need to rely on an additional reconstruction network.
>
> [1] Wonderland: Navigating 3D Scenes from a Single Image
>
> ---
>
> We hope our responses could address your questions regarding the architecture. We also welcome you to raise any further questions.

---

> > ### Comment · Reviewer_wnKK · 2025-11-25
> >
> > I thank the authors for resolving my concerns. I have no further questions at this time. I recommend acceptance of the paper.

---

> > > ### Author Response · Authors · 2025-11-25
> > >
> > > We thank you for your valuable comments and questions, which have significantly helped improve our manuscript.
> > >
> > > Additionally, we would like to kindly request that you reconsider the rating for a stronger recommendation, perhaps after you have fully evaluated your entire review batch.
> > >
> > > In any case, we sincerely appreciate your time and fully respect your decision.

---

> > > ### Author Response · Authors · 2025-11-25
> > >
> > > We sincerely appreciate you raising the rating. Your recognition has greatly encouraged us. Thank you again!

---

### Author Response · Authors · 2025-11-17

Dear Reviewers,

We would like to sincerely thank you for all your valuable suggestions, which have helped us make the paper more comprehensive and complete. In this revision, we have updated the following content:

1. Detailed workflow and pseudocode of few-step generation and cross-mode post-training in **[Appendix B, Lines 816-898]**.
2. More comparisons:
- Depth comparison with Director3D and Prometheus in **[Fig. 9, Lines 1065-1079]**.
- NVS comparison with ViewCrafter  in **[Fig. 10, Lines 1080-1010]**.
- Video quality assessment on Text-to-3D as follows:

**Table: Q-align VQA scores of rendered videos of different methods (higher is better)**

|  | T3Bench-200 | DL3DV-200 | WorldScore-200 |
| --- | --- | --- | --- |
| Director3D | 0.4841 | 0.6682 | 0.5388 |
| Promtheus | 0.4168 | 0.5088 | 0.5839 |
| Ours | **0.7676** | **0.8286** | **0.7219** |

3. An ablation study on image-to-3D scene generation task with WorldScore Benchmark as follows:

**Table: Ablation study of OOD co-training on WorldScore Benchmark**

|  | 3DConsistency | PhotometricConsistency | ObjectControl | ContentAlignment | StyleConsistency | SubjectiveQuality | Average |
| --- | --- | --- | --- | --- | --- | --- | --- |
| w/o OOD | **87.35** | **90.12** | 34.94 | 44.87 | **83.16** | 46.24 | 64.45 |
| Full model | 85.87 | 86.72 | **49.61** | **53.96** | 81.52 | **54.63** | **68.72** |

---

Regarding other concerns and questions you have raised, we have provided detailed and point-to-point responses under the corresponding comments.

We look forward to your feedback on our responses, and we also welcome you to raise new questions and engage in discussions.

Sincerely,

FlashWorld Authors

---

### Meta-Review · Area_Chair_wPog · 2025-12-28

**Summary:**

The reviewers recognized FlashWorld as a significant practical contribution to 3D scene generation, specifically noting the impressive 10–100× speedup over current baselines (achieving generation in ~9 seconds). While concerns on methodology details and experimental comparisons exist in initial reviews, they have been meticulously addressed by the authors in the rebuttal/discussion process. The consensus is that the work serves as a high-utility and technically sound framework that bridges the gap between speed and 3D consistency in 3D scene generation. I would recommend the acceptance of this paper. The authors are recommended to reflect the updates in the rebuttal, particularly the writing and pseudocode for Sec. 3.2, for the final revision.

**Reviewer Concerns:**

All major concerns raised by the reviewers were addressed and acknowledged during the discussion phase.
- Methodological Details (Sec 3.2): The authors successfully clarified the technical workflow by adding comprehensive discussions in Appendix B and providing the pseudocodes for the few-step generation process in Algorithm A and cross-mode post-training logic in Algorithm B.
- Empirical Validation of OOD: New quantitative ablations on the WorldScore Benchmark confirmed the benefits of the OOD co-training strategy.
- Video and Depth Performance: The authors emphasized quantitative scores in the original paper and added new scores/qualitative results on video and rendered depth.
- Comparisons with other baselines: The authors clarified the distinction between the proposed paper (for 3DGS) and ViewCrafter (for video).

**Reviewer Scores:**

The manuscript first received an initial review score of (6, 6, 6, 6). After the rebuttal/discussion and before the openreview incident, all reviewers acknowledged that their major concerns had been addressed, and two reviewers (wnKK & mRkm) raised their scores to 8, resulting in a score of (8, 6, 6, 8).

While reviewer N491 and m4Bg might raise their scores in later stages, as all reviewers have engaged in the discussion phase, I would approximate (8, 6, 6, 8) as the final score, leading to a clear acceptance of the paper. The paper also exhibits strong performance compared with concurrent works (Vista3D and Lyra), and thus, I would further recommend an oral presentation.

---

### Decision · Program_Chairs · 2026-01-26

Accept (Oral)